# FAITHFUL RULE LEARNING FOR TABULAR DATA CELL COMPLETION

## ABSTRACT

Tabular data cell completion aims to infer the correct constants that could fill a missing cell in a table row. While machine learning (ML) models have proven to be effective for this task, the limited interpretability restricts their applicability in trust-critical domains. In this paper, we develop two interpretable ML models to predict whether a candidate constant should fill the empty cell of an incomplete row by learning Datalog rules describing chain-like patterns of relations. Both models are *fully interpretable with formal guarantees*: we provide algorithms that take a model instance and extract an *equivalent* set of rules, in the sense that both the model and the rules produce the same output for any input table over a fixed relation schema. Furthermore, our models utilize different aggregation strategies to offer distinct trade-offs regarding expressive power and ease of rule extraction. Evaluations reveal that our models achieve state-of-the-art performance on tabular data cell completion with superior interpretability.

## 1 INTRODUCTION

Tabular data underlies many real-world datasets, from medical records to financial transactions. Tables, however, are often incomplete: missing cells arise due to collection errors, privacy restrictions, or unavailable information. This motivates the practical task to complete missing cells, with applications scenarios such as integrating heterogeneous Web tables, linking patient attributes across hospital databases, or reconciling missing entity fields in financial and organizational records. The task of tabular data cell completion aims to infer such missing values from the observed table entries (Zhang & Balog, 2019; Ahmadov et al., 2015; Yakout et al., 2012). Unlike classical imputation in machine learning (Sun et al., 2023; Van Buuren & Groothuis-Oudshoorn, 2011; Stekhoven & Bühlmann, 2012), cell completion does not assume a single ground-truth answer—multiple valid completions may exist, all of which are acceptable.

Early work approached this problem using statistical heuristics (Yakout et al., 2012), external resources such as knowledge bases (Zhang & Balog, 2019), specialized settings restricted to table positions such as headers (Ahmadov et al., 2015). More recently, machine learning methods have represented tabular data as hypergraphs, where nodes are cell values and hyperedges correspond to rows, and applied hypergraph neural networks (Yadati et al., 2019; Yadati, 2020). While effective, the predictions of these models are difficult to explain in a transparent and faithful manner, limiting trust and interpretability in sensitive domains. This is particularly critical in practical applications such as medical record completion or financial data reconciliation, where model transparency directly affects reliability and adoption. Our task also relates to neural rule learning methods developed for knowledge graph completion, such as Neural-LP (Yang et al., 2017) and DRUM (Sadeghian et al., 2019), though tabular data presents distinct challenges due to its multi-row, multi-column structure.

We address this challenge by drawing inspiration from faithful rule learning (Tena Cucala et al., 2022a;b; 2023; Wang et al., 2024a). Unlike feature-attribution explainability methods for tabular models (Lundberg & Lee, 2017), our goal is to design models for cell completion that not only achieve high predictive performance but also yield human-readable Datalog rules as explanations. Crucially, our approach provides a formal guarantee of faithfulness: every prediction made by the model can be derived from the extracted rules on the same input, and conversely, the rules derive exactly the model's predictions. For example, as illustrated in Figure 1, when predicting that a missing value is '*US*', the model may justify this via a rule connecting workplace and residency country. Such rules

| Name | Org. | Country |
|------|------|---------|
| *Emma* | *MIT* | *US* |
| ... | ... | ... |

| Name | Relation | Name |
|------|----------|------|
| *Emma* | *co-worker* | *Alice* |
| ... | ... | ... |

| Name | City | Country |
|------|------|---------|
| *Alice* | *Boston* | *US* |
| ... | ... | ... |
| *Emma* | *Boston* | **?** |

Figure 1: An example scenario of tabular data cell completion with multi-sourced Web tables about people's work and residency, where '?' denotes a missing value for the data cell.

offer a transparent view into the decision-making process of the model, fostering trust in the model's behavior.

Our contributions are: (i) Two novel rule learning models for cell completion: one based on sum aggregation (Section 3) and another based on max aggregation with reduced rule extraction complexity (Section 5); (ii) Faithful rule extraction algorithms for both models, with formal guarantees (Sections 4 and 5); and (iii) A comprehensive evaluation on standard benchmarks, showing competitive performance and improved explainability (Section 6).

## 2 BACKGROUND

**Tabular Data Cell Completion.** We follow standard database terminology. We assume a signature consisting of two disjoint countable sets of *predicates* and *constants*. Each predicate $P$ has an *arity* $|P|$ (its number of columns). A *fact* is an expression $P(c_1, \ldots, c_{|P|})$, where $P$ is a predicate and each $c_i$ is either a constant or the special *null* symbol '?'. Intuitively, a fact corresponds to a row in a table, and '?' denotes a missing value. A *table* over predicate $P$ is a finite set of facts over $P$. A database (instance) is a union of tables over predicates in a schema. If a fact, table, or database mentions '?', it is *incomplete*. For simplicity, we focus on the case where each fact contains at most one null value (extensions to multiple null values are discussed in Appendix F). For an incomplete database $\mathcal{D}$, a *completion* is another database obtained by replacing each null with one or more constants from $\mathcal{D}$. For example, a completion of the database in Figure 1 may replace '?' with the constant '*US*'. The task of *tabular data cell completion* is therefore: given $\mathcal{D}$, map each incomplete fact to the set of constants that replace its null value in $\mathcal{D}$'s completion (Yakout et al., 2012).

This task resembles *hypergraph link prediction* (Chen & Liu, 2023; Fatemi et al., 2020; Yadati et al., 2019; Wang et al., 2023a), since a database can be viewed as a hypergraph with constants as nodes and facts as hyperedges. However, unlike general link prediction (which predicts arbitrary facts), our task specifically fills missing cells in partially observed rows. Hypergraph methods can be adapted by removing incomplete facts, converting to a hypergraph, applying a model, and extracting predictions, but they lack direct mechanisms for handling incompleteness. The task is also related to *Programming by Example (PBE)* (Wang et al., 2017; Kolb et al., 2017), though in PBE users must supply input-output pairs and derivation knowledge, whereas here completions must be inferred automatically. We further discuss related work, including data-quality rule mining, in Appendix A.

**Datalog.** Datalog is a declarative rule-based language from databases and logic programming (Abiteboul et al., 1995), which we use to express symbolic rules explaining model predictions. A *term* is a variable or a constant. An *atom* is an expression of the form $P(t_1, \cdots, t_{|P|})$, where $P$ is a predicate and each $t_i$ with $1 \leq i \leq |P|$ is a term. An *inequality* is an expression as $t_1 \not\approx t_2$ where $t_1$ and $t_2$ are terms. A *literal* is an atom or an inequality. A *fact* is a variable-free atom, and a *dataset* is a finite set of facts. A (Datalog) *rule* is an expression of the form $H \leftarrow B_1 \wedge \cdots \wedge B_\ell$, where $\ell \geq 0$, $H$ is the *head atom*, and $B_i$ for $1 \leq i \leq \ell$ are *body literals*. A (Datalog) *program* is a finite set of rules.

For a mapping $\sigma$ from variables to terms, and for $\omega$ a term, a literal, or a conjunction of literals, $\omega\sigma$ replaces each variable $x$ in $\omega$ with $\sigma(x)$ if the latter is defined. Conjunctions of literals $\omega_1$ and $\omega_2$ are *isomorphic* if there exists a bijection $\sigma$ from the variables in $\omega_1$ to those in in $\omega_2$ such that $\omega_1\sigma$ and $\omega_2$ coincide. A set $S$ contains a conjunction of literals $\omega_1$ *up to isomorphism* if there exists $\omega_2 \in S$ such that $\omega_1$ and $\omega_2$ are isomorphic. Each rule $r$ defines an *immediate consequence* operator $T_r$. For a dataset $\mathcal{D}$, $T_r(\mathcal{D})$ is the smallest dataset containing $H\sigma$ for each mapping $\sigma$ from variables in $r$ to constants in $\mathcal{D}$ such that $B_i\sigma \in \mathcal{D}$ if $B_i$ is an atom, or $x\sigma \neq y\sigma$ if $B_i$ is an inequality of the form $x \not\approx y$, for each $1 \leq i \leq \ell$. Thus, inequalities are interpreted under the standard *Unique Name Assumption (UNA)*, that is, different constants refer to different objects. The immediate consequence

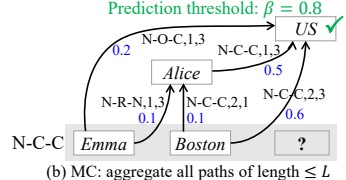
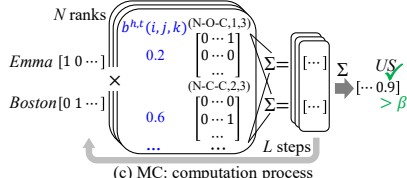

| Name | Org. | Country |
|------|------|---------|
| Emma | MIT | US |

| Name | Relation | Name |
|------|----------|------|
| Emma | co-worker | Alice |

| Name | City | Country |
|------|------|---------|
| Alice | Boston | US |
| Emma | Boston | ? |

(a) Input tabular data

(b) MC: aggregate all paths of length ≤ L

(c) MC: computation process

Figure 2: An illustration of our approach: the model aggregates all paths from existing constants (i.e., *Emma* and *Boston*) in the incomplete fact to a candidate constant (i.e., *US*) for completing it. Then it sums the weights of all paths and outputs constant *US* **if and only if** the sum surpasses the prediction threshold $\beta$. The values in blue are learnable parameters.

for a program $\mathcal{R}$ is defined as $T_{\mathcal{R}}(\mathcal{D}) = \bigcup_{r \in \mathcal{R}} T_r(\mathcal{D})$. The application of these operators to a dataset can only derive finitely many facts because there are finitely many constants in each input dataset.

**Tensors, Vectors, Matrices.** We consider $n$-dimensional *tensors* over $\mathbb{R}$. A *vector* is a 1-dimensional tensor, and a *matrix* is a 2-dimensional tensor. For an $n$-dimensional tensor $\mathbf{A}$, we use $\mathbf{A}(i_1, i_2, \cdots, i_n)$ to denote its element at position $(i_1, i_2, \cdots, i_n)$. For a list of tensors $\mathbf{T}_1, \cdots, \mathbf{T}_n$ of the same dimension, we use $\max_{1 \leq i \leq n} \mathbf{T}_i$ to denote their element-wise maximum. Besides, for matrices $\mathbf{M}$ of dimension $m \times n$ and $\mathbf{N}$ of dimension $n \times p$, the *max-product* of $\mathbf{M}$ and $\mathbf{N}$, written $\mathbf{M} \otimes \mathbf{N}$, is a matrix of dimension $m \times p$ where $\mathbf{M} \otimes \mathbf{N}(i, j) = \max_{1 \leq k \leq n} \mathbf{M}(i, k) \cdot \mathbf{N}(k, j)$ for $1 \leq i \leq m$ and $1 \leq j \leq p$.

# 3 MODEL FOR TABULAR DATA CELL COMPLETION

This section introduces the core idea of our model, presents its formal definition, demonstrates its application to input databases, and offers a formal interpretation of the function it realizes. We assume an arbitrary database schema $P_1, \ldots, P_\delta$ throughout the paper.

## 3.1 MODEL DEFINITION

Our method builds on rule-based link prediction models for graphs (Yang et al., 2017; Sadeghian et al., 2019; Tena Cucala et al., 2022b; Wang et al., 2024a), which learn to infer direct connections from path patterns between constants. We extend this concept to databases of any arity, where paths capture dependencies across richer relational structures.

**Definition 3.1.** A *path* of length $\ell \geq 1$ in a database $\mathcal{D}$ is a sequence $c_0, \cdots, c_\ell$ of (not necessarily distinct) constants such that for each $1 \leq i \leq \ell$, there exists a fact in $\mathcal{D}$ that mentions both $c_{i-1}$ and $c_i$ in distinct positions.

For example, given the database in Figure 2 (a), a path of length 2 is '*Emma, Alice, US*', as the database contains facts N-R-N(*Emma, co-worker, Alice*) and N-C-C(*Alice, Boston, US*)[1].

Our model predicts a missing constant (e.g., '*US*') by analyzing paths in $\mathcal{D}$ that connect this constant to other constants in the incomplete fact. For instance, in Figure 2, the aforementioned path from '*Emma*' to '*US*', represented in Figure 2 (b), can be used to predict that '*Emma*' lives in the '*US*'. Once the relevant paths have been identified, our model computes a weighted sum of their counts, where the weight of each path is based on its matching *path schemas*, as defined next.

**Definition 3.2.** A *path schema* of length $\ell \geq 1$ is a sequence of the form $(P_{r_1}, p_1, q_1)$, $(P_{r_2}, p_2, q_2), \cdots, (P_{r_\ell}, p_\ell, q_\ell)$, where $1 \leq r_i \leq \delta$ and $p_i, q_i$ are distinct positive integers with $1 \leq p_i, q_i \leq |P_{r_i}|$, for each $i \in \{1, \cdots, \ell\}$. A path $c_0, \cdots, c_\ell$ in $\mathcal{D}$ *matches* the path schema if there exist facts $\alpha_1, \cdots, \alpha_\ell$ in $\mathcal{D}$ such that $\alpha_i$ mentions predicate $P_{r_i}$ and has constant $c_{i-1}$ (resp. $c_i$) in position $p_i$ (resp. $q_i$), for each $1 \leq i \leq \ell$.

For example, in the database in Figure 2 (a), the path '*Emma, Alice, US*' matches the path schema '$(\text{N-R-N}, 1, 3), (\text{N-C-C}, 1, 3)$'. Note that a path can match several path schemas.

---

[1]We abbreviate the predicates as initials of column names, e.g., N-C-C for Name-City-Country.

Path schemas capture relevant properties for path analysis. Ideally, our model would assign an individual learnable parameter to each path schema, quantifying its influence in completing missing values over a specific predicate. The prediction process would involve identifying all paths from each constant in an incomplete fact to a candidate constant, weighting each path by its schema-specific parameters, and then aggregating the results. However, this approach would require a prohibitively large number of parameters, most of which would not be used in any given prediction. This would lead to training instability, particularly in deep architectures, where issues like vanishing or exploding gradients are exacerbated (Bengio et al., 1994). To address this, we adopt the solution in TensorLog (Cohen et al., 2020) and subsequent works like Neural-LP (Yang et al., 2017) and DRUM (Sadeghian et al., 2019), where path schema influences are computed as composite functions of a more compact set of parameters, improving both efficiency and training stability.

The following definition introduces the parameters of our model.

**Definition 3.3.** A *Multi-Chain (MC) model* of rank $N \geq 1$ and depth $L \geq 1$ is a tuple $(\mathbf{b}^{1,1}, \cdots, \mathbf{b}^{1,|P_1|}, \cdots, \mathbf{b}^{\delta,1}, \cdots, \mathbf{b}^{\delta,|P_\delta|}, \beta)$, where $\beta \in \mathbb{R}_{\geq 0}$ and each $\mathbf{b}^{h,t}$ is a tensor in $[0,1]^{N \times L \times K}$ with $K = \sum_{k=1}^{\delta} |P_k| \cdot (|P_k| - 1) + 1$.

The model's expressive power is controlled by two hyperparameters: rank $N$ and depth $L$. Rank $N$ indicates that the model simulates $N$ independent rank 1 models whose outputs are combined, while $L$ sets the maximum path length the model considers. Each tensor $\mathbf{b}^{h,t}$ corresponds to a possible predicate $P_h$ and a position $1 \leq t \leq |P_h|$; the third dimension $K$ of these tensors is one plus the total number of distinct triples $(P, p, q)$ that can appear in a path schema: indeed, for each of the $\delta$ possible distinct predicates $P_k$, there are exactly $|P_k| \cdot (|P_k| - 1)$ distinct pairs of distinct positions. The additional $+1$ corresponds to the empty fact $\top$, which our model will use to consider paths with length strictly less than $L$.

Let $\mathcal{M}$ be an MC model as in Definition 3.3 and let $P_h(\mathbf{c})$ be an incomplete fact in $\mathcal{D}$ with a null in position $t$. Model $\mathcal{M}$ predicts the constants replacing this null value as follows. First, let $c_1, c_2, \cdots, c_\epsilon$ be the constants in $\mathcal{D}$ in an arbitrary but fixed order. For each predicate $P_k$ and pair of distinct positions $p, q$ with $1 \leq p, q \leq |P_k|$, let $\mathbf{M}_{k,p \to q} \in \{0,1\}^{\epsilon \times \epsilon}$ be an adjacency matrix where $\mathbf{M}_{k,p \to q}(i,j) = 1$ if there is a fact in $\mathcal{D}$ over $P_k$ and constants $c_i$ and $c_j$ are on its $p$-th and $q$-th positions, respectively, otherwise $\mathbf{M}_{k,p \to q}(i,j) = 0$. For each rank $1 \leq i \leq N$ and constant $c_s \in \mathbf{c}$, the model encodes $c_s$ as a one-hot vector $\mathbf{v}_{c_s}^{i,0}$ where $\mathbf{v}_{c_s}^{i,0}(s) = 1$ and all other elements are 0. Then it iteratively computes the vector $\mathbf{v}_{c_s}^{i,j}$ for $1 \leq j \leq L$ as in Equation 1, where $\mathbf{d}_{k,p \to q} \in \mathbb{N}$ is the position of $(k, p, q)$ in the lexicographic order for $1 \leq k \leq \delta$, $1 \leq p, q \leq |P_k|$, and $p \neq q$ (see Appendix D for an analytical expression):

$$(\mathbf{v}_{c_s}^{i,j})^\intercal = (\mathbf{v}_{c_s}^{i,j-1})^\intercal \cdot \Big( \sum_{k=1}^{\delta} \sum_{1 \leq p,q \leq |P_k|,\, p \neq q} \mathbf{b}^{h,t}(i,j,\mathbf{d}_{k,p \to q}) \cdot \mathbf{M}_{k,p \to q} + \mathbf{b}^{h,t}(i,j,K) \Big). \quad (1)$$

The model then computes vector $\mathbf{v}_{\mathbf{c}}^{h,t} = \sum_{i=1}^{N} \sum_{c_s \in \mathbf{c}} \mathbf{v}_{c_s}^{i,L}$ by adding the results of all constants in $\mathbf{c}$ and all ranks. For each $1 \leq u \leq \epsilon$, constant $c_u$ replaces the null value iff $\mathbf{v}_{\mathbf{c}}^{h,t}(u) > \beta$.

## 3.2 MODEL INTERPRETATION

We next show that our model's operation implements the intuition outlined in Section 3.1, which involves calculating a weighted sum of all paths in database $\mathcal{D}$ that match each valid path schema up to a specified maximum length. Fix an MC model $\mathcal{M}$ as in Definition 3.3 with rank $N$ and depth $L$, and a database $\mathcal{D}$ over constants $c_1, \cdots, c_\epsilon$. Let $\Omega$ be the set of path schemas of length up to $L$ and $\top$. We provide an interpretation of the vectors $\mathbf{v}_{\mathbf{c}}^{h,t}$ as a weighted sum of relevant paths in $\mathcal{D}$.

**Lemma 3.4.** *Let $P_h(\mathbf{c})$ be an incomplete fact in $\mathcal{D}$ with a null value in position $1 \leq t \leq |P_h|$. Then, vector $\mathbf{v}_{\mathbf{c}}^{h,t}$ is equal to $\sum_{\omega \in \Omega} \mathtt{wt}_{\mathcal{M}}(\omega) \cdot \mathbf{v}_{\omega,\mathcal{D},\mathbf{c}}$, where $\mathtt{wt}_{\mathcal{M}} : \Omega \mapsto \mathbb{R}_{\geq 0}$ is a function that depends only on the parameters of $\mathcal{M}$, and $\mathbf{v}_{\omega,\mathcal{D},\mathbf{c}}$ is a non-negative vector of dimension $\epsilon$ defined as follows: (1) if $\omega = \top$, then $\mathbf{v}_{\omega,\mathcal{D},\mathbf{c}}(u)$ is the number of occurrences of $c_u$ in $\mathbf{c}$, for each $1 \leq u \leq \epsilon$; (2) if $\omega$ is a path schema of length $\ell \geq 1$, then $\mathbf{v}_{\omega,\mathcal{D},\mathbf{c}}(u)$ is the number of distinct paths in $\mathcal{D}$ connecting constants in $\mathbf{c}$ to $c_u$ and matching $\omega$, for each $1 \leq u \leq \epsilon$.*

Intuitively, when $\omega \neq \top$, each element $\mathbf{v}_{\omega,\mathcal{D},\mathbf{c}}(u)$ for $1 \leq u \leq \epsilon$ counts the number of distinct paths matching $\omega$ and connecting constants in $\mathbf{c}$ to $c_u$. The lemma holds because, for each $1 \leq i \leq N$,

constant $c_s$, and step $j$ in the computation of $\mathbf{v}_{c_s}^{i,L}$, element $\mathbf{v}_{c_s}^{i,j}(u)$ represents a weighted sum of all distinct paths from $c_s$ to $c_u$ in $\mathcal{D}$ of length $\leq j$. The (unweighted) sum of these paths can be factored out across all $1 \leq i \leq N$, leaving behind an expression that depends only on the parameters of $\mathcal{M}$. Equation 1 ensures that $\mathbf{v}_{c_s}^{i,j}(u)$ has the aforementioned meaning: the products of all $\mathbf{M}_{k,p\rightarrow q}$ with $\mathbf{v}_{c_s}^{i,j-1}$ essentially consider all possible extensions of paths of length $\leq j-1$ starting from $c_s$ (represented by $\mathbf{v}_{c_s}^{i,j-1}$) with an additional fact $\alpha$ or with $\top$, forming paths of length $\leq j$. The weight of each new path is obtained by multiplying the weight of the previous path by $\mathbf{b}^{h,t}(i,j,\mathtt{d}_{k,p\rightarrow q})$, where $P_k$ is the predicate of $\alpha$, $p$ is the position in $\alpha$ of the constant that links $\alpha$ with the previous path, and $q$ is the position of $c_u$ in $\alpha$. The last term $(\mathbf{v}_{c_s}^{i,j-1})^{\mathsf{T}} \cdot \mathbf{b}^{h,t}(i,j,K)$ corresponds to the case where each previous path is extended by $\top$ instead of a fact $\alpha$.

# 4 FAITHFUL RULE EXTRACTION FOR THE MC MODEL

In this section, we introduce an algorithm to extract a set of Datalog rules from an arbitrary MC model so that both the rules and the model generate the same outputs for each database. We also present a simpler algorithm to extract rules explaining specific model predictions.

## 4.1 MULTICHAIN RULES

To apply Datalog rules to databases, we follow the standard procedure and represent each database over the fixed database schema $\{P_i\}_{1\leq i\leq\delta}$ as a relational dataset. To represent incomplete facts, we introduce an *auxiliary predicate* $P_i^t$ of arity $|P_i| - 1$ for each $P_i$ and possible position $1 \leq t \leq |P_i|$ of the null value in a fact over $P_i$. We then represent each incomplete fact $P_i(c_1, \cdots, c_{t-1}, ?, c_{t+1}, \cdots, c_{|P_i|})$ as the fact $P_i^t(c_1, \cdots, c_{t-1}, c_{t+1}, \cdots, c_{|P_i|})$. For example, the incomplete fact in Figure 2 (a) is described by the fact $\texttt{N-C-C}^3(\textit{Emma}, \textit{Boston})$.

We next introduce *chain patterns*, which will describe path schemas in rule bodies.

**Definition 4.1.** A *chain pattern* $\Phi(x,y)$ of length $\ell \geq 1$ is an ordered conjunction of $\ell$ atoms where the $i$-th atom mentions *linking variables* $z_{i-1}$ and $z_i$ exactly once, with $z_0 = x$ and $z_\ell = y$, and no other variable occurs twice.

Each path schema corresponds to a chain pattern of the same length, where each element $(P_{r_i}, p_i, q_i)$ in the path schema corresponds to an atom with predicate $P_{r_i}$ and variables $z_{i-1}, z_i$ on the $p_i$-th and $q_i$-th positions, respectively, and fresh variables in the other positions. For example, consider the database schema in Figure 2 (a); a chain pattern describing the path schema $(\texttt{N-R-N}, 1, 3), (\texttt{N-C-C}, 1, 3)$ is $\texttt{N-R-N}(x, v, z_1) \wedge \texttt{N-C-C}(z_1, w, y)$.

As shown in Lemma 3.4, the MC model is capable of counting distinct paths between constants. To express this counting ability using rules, we use conjunctions of a special form:

**Definition 4.2.** A *multichain conjunction* for $x$ and $y$ with *core* chain pattern $\Phi(x,y)$ and cardinality $C \in \mathbb{N}$ is of the form $\bigwedge_{j=1}^{C} \Phi^j(x,y) \wedge \bigwedge_{1\leq j<j'\leq C} \left( \bigvee_{i=1}^{\ell-1} z_i^j \not\approx z_i^{j'} \right)$, where $\Phi^j(x,y)$ is obtained by replacing each $z_i$ in $\Phi(x,y)$ with $z_i^j$, and any variables other than $x$, $y$, and $z_i^j$ are pairwise distinct among all atoms in the conjunction.

A multichain conjunction for $x$ and $y$ is specified by a cardinality $C$ and a chain pattern $\Phi(x,y)$. Intuitively, it represents $C$ copies of the core $\Phi(x,y)$ where variables other than $x$ and $y$ are uniquely renamed in each copy. The inequalities ensure that, when the conjunction is grounded, no two copies ground all linking variables $z_1, \cdots, z_{\ell-1}$ in the same way (the disjunction ensures that they differ in the grounding of at least one variable). Hence, a multichain conjunction can be grounded in $\mathcal{D}$ if there are at least $C$ distinct paths in $\mathcal{D}$ with the same endpoints matching the path schema $\Phi(x,y)$.

For example, a multichain conjunction with core chain pattern $\texttt{N-R-N}(x,v,z) \wedge \texttt{N-C-C}(z,w,y)$ and cardinality 2 is $\texttt{N-R-N}(x,v_1,z_1) \wedge \texttt{N-C-C}(z_1,w_1,y) \wedge \texttt{N-R-N}(x,v_2,z_2) \wedge \texttt{N-C-C}(z_2,w_2,y) \wedge z_1 \not\approx z_2$, where $z_1 \not\approx z_2$ ensures that it can only be grounded to distinct paths with the same chain pattern.

**Definition 4.3.** A *Multichain (MC) rule* is of the form 2, where for each $1 \leq r \leq |P_h|$ with $r \neq t$, $x_r'$ is either $x_r$ or identical to $y$, and $\varphi_r(x_r', y)$ is a (possibly empty) conjunction of finitely many multichain conjunctions for $x_r'$ and $y$, with no variables in common other than $x_r'$ and $y$. Moreover,

---

**Algorithm 1:** Faithful Rule Extraction for an MC Model.

---

**Input:** An MC model $\mathcal{M}$, and a rule extraction threshold $\gamma$.
**Output:** A finite set $\mathcal{R}_\mathcal{M}$ of multichain rules.

1  $\mathcal{R}_\mathcal{M} := \emptyset$, $\Omega :=$ list of path schemas with $\ell \leq L$, ending with $\top$, **foreach** $\omega \in \Omega$ **do** $\mathtt{wt}(\omega) := 0$;

2  **foreach** $h, t \in \{(1,1), \cdots, (1, |P_1|), \cdots, (\delta, 1), \cdots, (\delta, |P_\delta|)\}$ **do**

3     **foreach** $[d_1, \cdots, d_L]$ with $d_i \in \{1, \cdots, K\}$ **do**

4        $[d'_1, \cdots, d'_\ell] :=$ remove all occurrences of $K$ from $[d_1, \cdots, d_L]$;

5        **foreach** $j \in \{1, \cdots, \ell\}$ **do**

6           $(k_j, p_j, q_j) :=$ the triple corresponding to $d'_j$;

7        **if** $\ell \geq 1$ **then** $\omega = (k_1, p_1, q_1), \cdots, (k_\ell, p_\ell, q_\ell)$ **else** $\omega := \top$ ;

8        $\mathtt{wt}(\omega) := \mathtt{wt}(\omega) + \sum_{i=1}^N \prod_{j=1}^L \mathbf{b}^{h,t}(i, j, d_j)$;

9     **foreach** $i \in \{1, \cdots, |\Omega| - 1\}$ **do**

10        **if** $\mathtt{wt}(\Omega(i)) = 0$ **then** $\mathring{C}_i := 0$ **else if** $|\Omega(i)| = 1$ **then** $\mathring{C}_i := 1$ **else** $\mathring{C}_i := \lfloor \frac{\beta}{\mathtt{wt}(\Omega(i))} \rfloor + 1$;

11     $\Theta := \emptyset$;

12     **foreach** $[C_1, \cdots, C_{|\Omega|-1}]$ where each $C_i \in \{0, \cdots, \mathring{C}_i\}$ **do**

13        $\varphi := \bigwedge_{i=1}^{|\Omega|-1} \phi_i$, where if $C_i = 0$ then $\phi_i := \top$ else $\phi_i :=$ an MC conjunction for $x$ and $y$ with cardinality $C_i$ and the chain pattern for $\Omega(i)$ as core;

14        $\Theta := \Theta \cup \{\varphi, \varphi\{x \mapsto y\}\}$;

15        $\mathtt{wt}'(\varphi) := \sum_{i=1}^{|\Omega|-1} C_i \cdot \mathtt{wt}(\Omega(i))$;   $\mathtt{wt}'(\varphi\{x \mapsto y\}) := \mathtt{wt}'(\varphi) + \mathtt{wt}(\top)$;

16     **foreach** $[\varphi_1, \cdots, \varphi_{t-1}, \varphi_{t+1}, \cdots, \varphi_{|P_h|}]$ with $\varphi_r \in \Theta$ **do**

17        **if** $\sum_{1 \leq r \leq |P_h|, r \neq t} \mathtt{wt}'(\varphi_r) \leq \gamma$ **then continue**;

18        $H := P_h(x_1, \cdots, x_{t-1}, y, x_{t+1}, \cdots, x_{|P_h|})\{x_r \mapsto y \text{ foreach } \varphi_r \text{ not mentioning } x\}$;

19        $A := P_h^t(x_1, \cdots, x_{t-1}, x_{t+1}, \cdots, x_{|P_h|})\{x_r \mapsto y \text{ foreach } \varphi_r \text{ not mentioning } x\}$;

20        $\mathcal{R}_\mathcal{M} := \mathcal{R}_\mathcal{M} \cup \big\{ H \leftarrow A \wedge \bigwedge_{1 \leq r \leq |P_h|, r \neq t} \varphi_r \{x \mapsto x_r \text{ if } x_r \text{ appears in } H\} \big\}$;

21  **return** $\mathcal{R}_\mathcal{M}$;

---

at least one $\varphi_r(x'_r, y)$ must be non-empty if no $x'_r$ is identical to $y$.

$$P_h(x'_1, \cdots, x'_{t-1}, y, x'_{t+1}, \cdots, x'_{|P_h|}) \leftarrow P_h^t(x'_1, \cdots, x'_{t-1}, x'_{t+1}, \cdots, x'_{|P_h|}) \wedge \bigwedge_{\substack{1 \leq r \leq |P_h| \\ r \neq t}} \varphi_r(x'_r, y) \quad (2)$$

The body of a rule of the form 2 matches a dataset $\mathcal{D}$ if the corresponding database has an incomplete fact of the form $P_h(c_{s_1}, \cdots, c_{s_{t-1}}, ?, c_{s_{t+1}}, \cdots, c_{s_{|P_h|}})$, as per the first body atom, and if it contains enough paths from the constants $c_{s_r}$ to some constant $c$ matching the path schemas captured by the conjunctions in the relevant $\varphi_r(x'_r, y)$. Then it derives $P_h(c_{s_1}, \cdots, c_{s_{t-1}}, c, c_{s_{t+1}}, \cdots, c_{s_{|P_h|}})$, replacing the null value in the incomplete fact by $c$.

## 4.2 Faithful Rule Extraction for an MC Model

We consider a set of rules $\mathcal{R}$ is *faithful* to an MC model $\mathcal{M}$ if, for any database $\mathcal{D}$, the result of applying $T_\mathcal{R}$ to $\mathcal{D}$ and then transforming the derived facts back into a database yields the same facts as directly applying $\mathcal{M}$ to $\mathcal{D}$ (Tena Cucala et al., 2022b; Wang et al., 2024a). Our goal in this section is to extract a faithful program $\mathcal{R}_\mathcal{M}$ from an arbitrary MC model $\mathcal{M}$.

While MC conjunctions are limited by their cardinality in counting paths, MC models can count arbitrarily. This gap is addressed by a crucial property: for each path schema $\omega$ with positive weight, there exists a cutoff value $\mathring{C}(\omega)$ beyond which additional matching paths do not affect the model's output. In particular, $\mathring{C}(\omega)$ is the smallest natural number that surpasses the model's threshold when multiplied by the model's weight for $\omega$. Our definitions ensure that a model completes a fact with constant $c$ if the number of paths matching $\omega$ from other constants in the fact to $c$ is at least $\mathring{C}(\omega)$, regardless of other possible paths between these constants. Hence, beyond $\mathring{C}(\omega)$, the output of the model is invariant to the number of paths matching $\omega$. Thus, to represent an MC model, we only need to consider finitely many rules where the cardinalities in MC conjunctions for path schema $\omega$ are at most $\mathring{C}(\omega)$.

Algorithm 1 describes the computation of a faithful program for an MC model. It initializes the program as an empty set, creates a list $\Omega$ of path schemas with length $\ell \leq L$ and $\top$, and initializes

a function $\mathtt{wt}(\cdot)$ to record their weights (line 1). Each element of $\Omega$ other than $\top$ corresponds to a chain pattern. The algorithm then iterates over predicates $P_h$ and positions $1 \le t \le |P_h|$ of the null value (line 2). Each iteration computes the weight $\mathtt{wt}(\omega)$ for each $\omega$ in $\Omega$ (lines 3–8). Then, for each path schema $\Omega(i)$, it computes the cardinality upper bound $\mathring{C}_i = \mathring{C}(\Omega(i))$ for the chain pattern corresponding to path schema $\Omega(i)$ (lines 9–10). Next, $\Theta$ is initialized as empty (line 11) to store each possible conjunction $\varphi$ comprised of at most one multichain conjunction per path schema $\omega$ in $\Omega$, with core $\omega$ and cardinality less or equal to the upper bound $\mathring{C}(\omega)$ (lines 12–13). Besides, $\Theta$ also contains copies of these conjunctions where $x$ is replaced by $y$, corresponding to chains that start and end in the same constant (line 14). The algorithm computes a weight $\mathtt{wt}'(\varphi)$ for each $\varphi \in \Theta$ by summing the weights $\mathtt{wt}(\omega)$ of the core $\omega$ of each multichain conjunction in $\varphi$ multiplied by its cardinality (line 15). Finally, the algorithm considers all possible ways of selecting one $\varphi_r \in \Theta$ for each $1 \le r \le |P_h|$ and $r \ne t$ (lines 16–20), and it computes a score for their conjunction by aggregating the weights $\mathtt{wt}'(\varphi_r)$ (line 17). If the score exceeds the rule extraction threshold $\gamma$, the rule constructed in lines 18–20 is added to the program.

**Theorem 4.4.** *The program $\mathcal{R}_{\mathcal{M}}$ extracted by Algorithm 1 is faithful to $\mathcal{M}$ when $\gamma = \beta$. Algorithm 1 terminates in $\mathcal{O}\left(C^{\delta^L \nu^{2L}}\right)$ steps with $\nu = \max_{1 \le k \le \delta} |P_k|$, $C = \max_{1 \le i \le |\Omega|-1} \mathring{C}_i$.*

### 4.3 Faithful Rule Extraction over a Specific Database

Although extracting a faithful program from an MC model can be computationally expensive, we show that a small subset of this program can be efficiently extracted to explain the predictions of the model for a specific database. To this end, we devise an algorithm that takes both a model and a dataset (corresponding to a database) as input, and extracts a single rule for each individual prediction of the model for that dataset. Specifically, for each prediction, the algorithm identifies all paths of length at most $L$ connecting the incomplete fact's constants to the predicted constant. Then it generates a multichain rule by using each matched path schema as a rule core and setting the cardinality to the schema's number of matches, bounded by the corresponding cutoff value. Finally, it returns the union of all these rules. The full algorithm is given in Appendix B.

**Theorem 4.5.** *For MC model $\mathcal{M}$ and dataset $\mathcal{D}$, a program $\mathcal{R}_{\mathcal{M},\mathcal{D}} \subseteq \mathcal{R}_{\mathcal{M}}$ that returns the same output as $\mathcal{M}$ on $\mathcal{D}$ can be obtained with worst time complexity $\mathcal{O}\left(\delta^L \cdot \nu^{2L} \cdot \epsilon^{L+\nu}\cdot\right)$, for $\nu = \max_{1 \le k \le \delta} |P_k|$ and $\epsilon$ the number of distinct constants in $\mathcal{D}$.*

Unlike in Theorem 4.4, the complexity of rule extraction here depends on the graph structure of $\mathcal{D}$ and the size of the result obtained by applying $\mathcal{M}$ to $\mathcal{D}$. In practice, datasets are often sparse (Appendix G). $\mathcal{R}_{\mathcal{M},\mathcal{D}}$ is typically computed in just a few minutes (Appendix H.4).

## 5 Model Variant with Simplified Rule Extraction

Recall that the MC model predicts that constant $c$ completes a fact by aggregating all paths in the database from the incomplete fact's constants to $c$. However, in practice, this approach can sometimes be too sensitive to the noise produced by irrelevant paths with small weights. To address this, we propose MC-max, a variant of the model that uses only the highest-weight path from a relevant constant to the target. While less expressive (see Appendix E), the MC-max model enables more efficient rule extraction with lower complexity.

The MC-max model simplifies the original MC approach by replacing path counting with binary path existence checks. This is achieved through three key modifications to the MC computation: first, pushing vector terms into operators; second, replacing all summation operators with max operators (including matrix products becoming max-products); and third, using max operations for aggregating intermediate results. These changes preserve the model's ability to detect paths while eliminating its counting capability.

**Definition 5.1.** An MC-max model has the same form as an MC model, with Equation 1 replaced by Equation 3 and $\mathbf{v}_{\mathbf{c}}^{h,t} = \max_{1 \le i \le N, c_s \in \mathbf{c}} \mathbf{v}_{c_s}^{i,L}$.

$$(\mathbf{v}_{c_s}^{i,j})^{\intercal} = \max_{\substack{1 \le k \le \delta \\ 1 \le p,q \le |P_k|, \, p \ne q}} \left\{ \mathbf{b}^{h,t}(i,j,K) \cdot (\mathbf{v}_{c_s}^{i,j-1})^{\intercal}, \mathbf{b}^{h,t}(i,j,\mathtt{d}_{k,p \to q}) \cdot (\mathbf{v}_{c_s}^{i,j-1})^{\intercal} \otimes \mathbf{M}_{k,p \to q} \right\}. \quad (3)$$

---

**Algorithm 2:** Faithful Rule Extraction for an MC-max Model.

---

**Input:** An MC-max model $\mathcal{M}$, and a rule extraction threshold $\gamma$.
**Output:** A Datalog (without inequality) program $\mathcal{R}_\mathcal{M}$.

1   $\mathcal{R}_\mathcal{M} := \emptyset$;
2   **foreach** $h, t \in \{(1,1), \cdots, (1, |P_1|), \cdots, (\delta, |P_\delta|)\}$ and $i \in \{1, \cdots, N\}$ **do**
3     $\mathcal{S} := \emptyset$; $\mathcal{S}' := \{(1, [])\}$;
4     **foreach** $j \in \{1, \cdots, L\}$ **do**
5       $\mathcal{S} := \mathcal{S}'$; $\mathcal{S}' := \emptyset$;
6       **foreach** $(s, [k]) \in \mathcal{S}$ and $k' \in \{1, \cdots, K\}$ **do**
7         **if** $s \cdot \mathbf{b}^{h,t}(i, j, k') > \gamma$ **then** $\mathcal{S}' = \mathcal{S}' \cup \{(s \cdot \mathbf{b}^{h,t}(i, j, k'), [k, k'])\}$;
8     **foreach** $(s, [k_1, \cdots, k_L]) \in \mathcal{S}'$ **do**
9       $[k'_1, \cdots, k'_\ell] :=$ remove all occurrences of $K$ from $[k_1, \cdots, k_L]$;
10      **foreach** $j \in \{1, \cdots, \ell\}$ **do**
11        $(k, p, q) :=$ the triple corresponding to $k'_j$; $\varphi_j :=$ atom with predicate $P_k$, variables $z_{j-1}, z_j$
          on the $p$-th, $q$-th positions, and fresh variables on the other positions, resp.;
12      **foreach** $r \in \{1, \cdots, t-1, t+1, \cdots, |P_h|\}$ **do**
13        $H := P_h(x_1, \cdots, x_{t-1}, y, x_{t+1}, \cdots, x_{|P_h|})$, $A := P_h^t(x_1, \cdots, x_{t-1}, x_{t+1}, \cdots, x_{|P_h|})$;
14        **if** $\ell \geq 1$ **then** $\mathcal{R}_\mathcal{M} := \mathcal{R}_\mathcal{M} \cup \{H \leftarrow A \wedge \bigwedge_{j=1}^\ell \varphi_j \{z_0 \mapsto x_r, z_\ell \mapsto y\}\}$ **else**
          $\mathcal{R}_\mathcal{M} := \mathcal{R}_\mathcal{M} \cup \{H\{x_r \mapsto y\} \leftarrow A\{x_r \mapsto y\}\}$;
15   **return** $\mathcal{R}_\mathcal{M}$;

---

The complexity of rule extraction from an MC model stems from enumerating all possible path schema matches. By eliminating path counting, MC-max simplifies rule extraction while preserving core functionality. Algorithm 2 describes rule extraction for MC-max. It iterates over all predicates $P_h$, position $t$, and rank $i$ (line 2). Each iteration expands the path schema and updates the weight by multiplying the value $\mathbf{b}^{h,t}(i, j, k)$ accordingly (lines 4–7). Since elements of $\mathbf{b}^{h,t}$ are between 0 and 1, the updated weight value in each step decreases monotonically. Hence, we can prune the search space by comparing the current weight with the threshold in each step (line 7). For each path schema whose weight exceeds the threshold (line 8), the algorithm constructs the corresponding chain pattern (lines 9–11), enumerates all $x_r$ (line 12), and adds the rule to the output program (lines 13–14).

**Theorem 5.2.** *The program $\mathcal{R}_\mathcal{M}$ extracted by Algorithm 2 is faithful to the input MC-max model $\mathcal{M}$ when $\gamma = \beta$. Algorithm 2 terminates in $\mathcal{O}\left(\delta^L \cdot \nu^{2L}\right)$ steps with $\nu = \max_{1 \leq k \leq \delta} |P_k|$.*

## 6   Evaluation

**Datasets and Baselines.** We used the relational tabular datasets WP-IND, JF-IND, and MFB-IND (Yadati, 2020) under the inductive setting, where constants in the test sets may not appear in training. We also used FB-AUTO (Fatemi et al., 2020) under the transductive setting, where all test-time constants are seen during training. We also evaluated our models on binary inductive datasets from Teru et al. (2020), based on FB15k-237 (Toutanova & Chen, 2015), NELL-995 (Xiong et al., 2017), and WN18RR (Dettmers et al., 2018), with original splits. We use HyperGCN (Yadati et al., 2019), GMPNN (Yadati, 2020) and two LLM-based methods, Chain-of-Thought (CoT) (Wei et al., 2022) with GPT-5 mini (OpenAI, 2025), TabLLM (Hegselmann et al., 2023) as baselines (more details in Appendix G).

**Tabular Data Cell Completion.** Table 1 reports Precision (P), Recall (R), Accuracy (Acc), Area Under the precision-recall Curve (AUC) and F1 score. In general, MC and MC-max outperformed baseline models, with MC-max slightly surpassing MC despite its limited expressivity. This supports our hypothesis that aggregating all paths may introduce noise. GMPNN models achieved higher recall on most datasets, which we attribute to their use of randomly selected negative examples for training augmentation. LLM-based methods exhibit unbalanced behavior: few-shot prompting results are often overly conservative, resulting in low recall, while fine-tuned models become overly optimistic, producing many false positives. Table 2 shows that MC and MC-max models achieve similar overall performance on binary datasets, with complementary strengths: MC has better precision and accuracy, while MC-max yields higher recall. These results suggest MC-max's practical applicability despite its simpler design. (More result analysis in Appendix H.)

Table 1: Results (%) of Inductive and Transductive Tabular Data Cell Completion.

| | WP-IND | | | | | JF-IND | | | | | MFB-IND | | | | | FB-AUTO | | | | |
|---|---|---|---|---|---|---|---|---|---|---|---|---|---|---|---|---|---|---|---|---|
| | P | R | Acc | AUC | F1 | P | R | Acc | AUC | F1 | P | R | Acc | AUC | F1 | P | R | Acc | AUC | F1 |
| HyperGCN | 54.5 | 38.4 | 53.2 | 46.5 | 45.0 | 58.1 | 30.9 | 54.3 | 44.5 | 40.4 | 61.8 | 29.0 | 55.5 | 45.4 | 39.4 | 67.4 | 76.4 | 69.7 | 71.9 | 71.6 |
| GMPNN-sum | 50.0 | 42.2 | 50.0 | 38.4 | 45.8 | 51.8 | 60.0 | 52.0 | 47.5 | 55.6 | 68.4 | 73.4 | 69.7 | 65.6 | 70.8 | 55.4 | 58.6 | 55.7 | 55.7 | 57.0 |
| GMPNN-mean | 52.4 | **94.5** | 54.3 | 60.4 | 67.4 | 53.1 | **92.7** | 55.5 | 58.9 | 67.5 | 69.1 | 81.1 | 72.4 | 72.4 | 74.6 | 58.1 | 85.6 | 61.9 | 71.9 | 69.2 |
| GMPNN-max | 50.3 | 78.3 | 50.5 | 53.6 | 61.3 | 53.2 | 87.7 | 55.2 | 64.8 | 66.2 | 80.4 | 92.4 | 84.9 | 89.2 | 86.0 | 60.3 | 94.9 | 66.2 | 77.4 | 73.8 |
| CoT (GPT-5 mini, 3 shot) | 87.1 | 29.2 | 62.0 | - | 43.8 | 96.4 | 40.1 | 69.3 | - | 56.6 | 84.6 | 11.1 | 54.5 | - | 19.6 | 90.5 | 7.2 | 50.6 | - | 13.3 |
| CoT (GPT-5 mini, 5 shot) | 88.0 | 28.9 | 62.0 | - | 43.5 | **96.7** | 43.1 | 70.8 | - | 59.6 | 85.7 | 12.1 | 55.1 | - | 21.2 | 92.0 | 8.7 | 51.4 | - | 15.9 |
| TabLLM (Llama 3.1) | 25.3 | 89.2 | 30.2 | - | 39.5 | 24.4 | 85.7 | 39.1 | - | 38.0 | 18.4 | 87.5 | 33.3 | - | 30.4 | 34.9 | **95.4** | 36.8 | - | 51.1 |
| MC ($L = 2, N = 1$) | 79.6 | 49.8 | 68.5 | 64.5 | 61.2 | 63.6 | 57.3 | 62.2 | 57.0 | 60.3 | **91.7** | 86.1 | 89.1 | 93.3 | 88.8 | 92.4 | 76.6 | 85.1 | 85.8 | 83.7 |
| MC ($L = 2, N = 2$) | 80.6 | 50.2 | 69.1 | 62.7 | 61.9 | 62.7 | 57.3 | 61.5 | 56.6 | 59.9 | 91.3 | 85.7 | 88.7 | 93.0 | 88.4 | 91.1 | 75.1 | 83.9 | 84.9 | 82.3 |
| MC ($L = 2, N = 3$) | 83.2 | 49.8 | 69.9 | 63.5 | 62.3 | 60.4 | 58.2 | 59.9 | 56.4 | 59.3 | 91.4 | 86.3 | 89.1 | **93.4** | 88.8 | 92.5 | 76.4 | 85.1 | 85.6 | 83.6 |
| MC-max ($L = 2, N = 1$) | 88.4 | 47.0 | 70.4 | 66.7 | 61.4 | 79.2 | 55.5 | 70.4 | 66.8 | 65.2 | 88.3 | 92.3 | 90.1 | 92.0 | 90.3 | 95.3 | 78.1 | 87.1 | 86.7 | 85.9 |
| MC-max ($L = 2, N = 2$) | **88.9** | 47.5 | 70.8 | 67.2 | 61.9 | 80.0 | 54.5 | 70.4 | 67.6 | 64.9 | 88.6 | **94.3** | **91.1** | 92.0 | **91.3** | **97.4** | 78.8 | **88.3** | 86.7 | **87.1** |
| MC-max ($L = 2, N = 3$) | 87.4 | 49.3 | 71.1 | 66.9 | 63.1 | 78.9 | 52.7 | 69.2 | 67.3 | 63.2 | 89.5 | 91.9 | 90.6 | 93.3 | 90.7 | 95.1 | 75.4 | 85.7 | 85.8 | 84.1 |
| MC-max ($L = 3, N = 1$) | 80.8 | 56.8 | 71.7 | **71.1** | 66.8 | 83.2 | 60.9 | **74.3** | 68.0 | **70.3** | 84.7 | 72.8 | 79.8 | 80.7 | 78.3 | 90.0 | 81.2 | 86.0 | **91.1** | 85.3 |
| MC-max ($L = 3, N = 2$) | 84.8 | 52.1 | 71.3 | 70.8 | 64.5 | 74.4 | 59.5 | 69.5 | **68.9** | 66.2 | 88.3 | 82.6 | 85.8 | 90.5 | 85.4 | 90.8 | 77.6 | 84.6 | 88.1 | 83.7 |
| MC-max ($L = 3, N = 3$) | 76.8 | 62.6 | **71.8** | 69.6 | **68.9** | 78.3 | 57.3 | 70.6 | 65.0 | 66.1 | 89.7 | 89.8 | 89.7 | 92.1 | 89.7 | 95.1 | 79.2 | 87.5 | 89.5 | 86.4 |

Table 2: Results (%) of MC (M) and MC-max (m) on Binary Datasets.

| | | P | | R | | Acc | | AUC | | F1 | |
|---|---|---|---|---|---|---|---|---|---|---|---|
| | | M | m | M | m | M | m | M | m | M | m |
| FB15k-237 | V1 | **47.4** | 47.2 | 44.9 | 44.9 | **47.6** | 47.3 | **51.6** | 51.0 | **46.1** | 46.0 |
| | V2 | **56.9** | 54.0 | 54.1 | **56.1** | **56.6** | 54.2 | **62.6** | 62.2 | **55.5** | 55.0 |
| | V3 | 65.1 | **65.5** | 46.0 | **46.2** | 60.7 | **60.9** | **60.9** | 60.5 | 53.9 | **54.2** |
| | V4 | **77.4** | 72.2 | 46.6 | **50.8** | **66.5** | 65.6 | 65.3 | 65.3 | 58.2 | **59.7** |
| NELL-995 | V1 | 97.5 | **100.0** | **76.5** | 68.5 | **87.2** | 84.2 | **80.4** | 79.7 | **85.7** | 81.3 |
| | V2 | **68.5** | 64.4 | 68.0 | **75.2** | **68.4** | 66.9 | 75.0 | **77.0** | 68.2 | **69.4** |
| | V3 | **66.4** | 62.9 | 76.2 | **78.1** | **68.8** | 66.1 | 80.2 | **81.6** | **71.0** | 69.7 |
| | V4 | **67.1** | 66.6 | 76.4 | **77.5** | **69.5** | 69.4 | 80.9 | 80.9 | 71.4 | **71.7** |
| WN18RR | V1 | **96.0** | 84.7 | 63.0 | **64.6** | **80.2** | 76.5 | 75.1 | **76.0** | **76.1** | 73.3 |
| | V2 | **92.5** | 89.9 | 61.9 | **62.2** | **78.5** | 77.6 | 73.0 | **73.4** | **74.2** | 73.5 |
| | V3 | **98.5** | 98.3 | 28.0 | 28.0 | 63.8 | 63.8 | **41.4** | 40.6 | 43.6 | 43.6 |
| | V4 | **95.3** | 93.4 | 59.4 | **59.6** | **78.3** | 77.7 | 71.4 | **71.7** | **73.2** | 72.7 |

Table 3: Rule Extraction Time (s) for MC-max.

| | | $N = 1$ | $N = 2$ | $N = 3$ |
|---|---|---|---|---|
| WP-IND | | 1.375 | 2.251 | 2.371 |
| JF-IND | | 0.777 | 1.205 | 1.323 |
| MFB-IND | | 0.025 | 0.030 | 0.039 |
| FB-AUTO | | 0.005 | 0.005 | 0.007 |
| FB15k-237 | V1 | 54.87 | 100.9 | 109.6 |
| | V2 | 71.75 | 138.0 | 159.8 |
| | V3 | 39.30 | 84.83 | 135.8 |
| | V4 | 78.28 | 95.78 | 142.1 |
| NELL-995 | V1 | 0.006 | 0.009 | 0.009 |
| | V2 | 4.988 | 5.799 | 10.74 |
| | V3 | 21.90 | 40.82 | 66.59 |
| | V4 | 3.634 | 6.998 | 10.51 |
| WN18RR | V1 | 0.001 | 0.001 | 0.002 |
| | V2 | 0.002 | 0.002 | 0.003 |
| | V3 | 0.001 | 0.002 | 0.003 |
| | V4 | 0.001 | 0.001 | 0.002 |

**Rule Extraction.** We implemented Algorithm 2 for faithful rule extraction for MC-max. Table 3 reports its runtime for each dataset, which was under 3 minutes in all cases. We also implemented Section 4.3's database-specific rule extraction algorithm for MC. We recorded rule extraction time for every 100 facts predicted by MC on each test dataset, which finished within 5 minutes in all cases (Appendix H.4). These results validate the feasibility of our approaches.

Table 4 reports the size of the extracted rule sets under different thresholds. Overall, the number of rules ranges from tens to a few thousand, with an average of at most 10–20 rules per head atom. This indicates the burden of reading and understanding is acceptable for real-world users. Intuitively, a higher rank (N) introduces greater diversity in the learned rules. By adjusting $N$, $L$ and $\gamma$, users could achieve a trade-off between rule compactness and explanatory richness.

Table 4: Number of Rules Extracted under Different $\gamma$.

| | WP-IND | | | | | JF-IND | | | | | MFB-IND | | | | | FB-AUTO | | | | |
|---|---|---|---|---|---|---|---|---|---|---|---|---|---|---|---|---|---|---|---|---|
| $\gamma$ | 0.8 | 0.4 | 0.2 | 0.1 | 0.01 | 0.8 | 0.4 | 0.2 | 0.1 | 0.01 | 0.8 | 0.4 | 0.2 | 0.1 | 0.01 | 0.8 | 0.4 | 0.2 | 0.1 | 0.01 |
| MC-max ($L = 2, N = 1$) | 7 | 34 | 59 | 93 | 885 | 6 | 25 | 43 | 77 | 664 | 28 | 36 | 42 | 47 | 78 | 1 | 8 | 14 | 29 | 223 |
| MC-max ($L = 2, N = 2$) | 7 | 37 | 77 | 121 | 1,614 | 4 | 37 | 64 | 126 | 1,877 | 44 | 68 | 84 | 104 | 206 | 10 | 22 | 41 | 72 | 359 |
| MC-max ($L = 2, N = 3$) | 9 | 55 | 102 | 187 | 3,010 | 8 | 48 | 91 | 179 | 3,131 | 61 | 98 | 125 | 156 | 326 | 6 | 22 | 51 | 102 | 562 |
| MC-max ($L = 3, N = 1$) | 1 | 6 | 16 | 27 | 231 | 1 | 10 | 20 | 38 | 432 | 17 | 21 | 39 | 59 | 248 | 1 | 6 | 19 | 25 | 129 |
| MC-max ($L = 3, N = 2$) | 2 | 13 | 25 | 50 | 582 | 1 | 11 | 24 | 59 | 835 | 34 | 55 | 89 | 112 | 353 | 2 | 8 | 22 | 42 | 440 |
| MC-max ($L = 3, N = 3$) | 3 | 21 | 36 | 56 | 751 | 31 | 98 | 119 | 160 | 1,485 | 61 | 88 | 120 | 159 | 472 | 2 | 17 | 33 | 58 | 504 |

Table 5 presents examples of top-weighted rules extracted by MC-max from the WP-IND, JF-IND, and MFB-IND datasets. Most extracted rules demonstrate binary relationships between constants, such as the first two rules from WP-IND. The model also captured symmetrical relationships, exemplified by the third rule from WP-IND, as well as transitive relationships, such as the first rule

from MFB-IND. Additionally, it exhibits the ability to 'align' specific positions across different predicates. For example, in the third rule extracted from JF-IND, it aligns the positions of `Team` and `Player` across different predicates `Player-Event-Team` and `Team-Player-Event`. More extracted rules are provided in Appendix H.5 with analysis.

Table 5: Example Rules Learned by MC-max from Tabular Datasets.

| | |
|---|---|
| **WP-IND** | Deceased-Place-Country$(x_1,x,y) \leftarrow$ Deceased-Place-Country$^3(x_1,x) \wedge$ Deceased-Place-Country$(w_1,x,y)$
Member-Membership-Role$(x_1,x,y) \leftarrow$ Member-Membership-Role$^3(x_1,x) \wedge$ Member-Membership-Role$(w_1,x,y)$
Person-Sibling-Kinship$(x,y,x_2) \leftarrow$ Person-Sibling-Kinship$^2(x,x_2) \wedge$ Person-Sibling-Kinship$(y,x,w_1)$
Politician-Position-Predecessor-Successor$(x,x_2,y,x_3) \leftarrow$
    Politician-Position-Predecessor-Successor$^3(x,x_2,x_3) \wedge$ Politician-Position-Predecessor-Successor$(y,w_1,w_2,x)$ |
| **JF-IND** | Country-Player-Olympics$(y,x,x_2) \leftarrow$ Country-Player-Olympics$^1(x,x_2) \wedge$ Country-Player-Olympics$(y,x,w_1)$
Player-Event-Player$(x_1,y,x) \leftarrow$ Player-Event-Player$^2(x_1,x) \wedge$ Player-Event-Player$(x,y,w_1)$
Player-Event-Team$(x,x_2,y) \leftarrow$ Player-Event-Team$^2(x,x_2) \wedge$ Team-Player-Event$(y,x,w_1)$
Team-Player-Event$(x,x_2,y) \leftarrow$ Team-Player-Event$^3(x,x_2) \wedge$ Team-Player-Event$(x,w_1,y)$ |
| **MFB-IND** | Ethnicity-Language-Person$(x_1,y,x) \leftarrow$
    Ethnicity-Language-Person$^2(x_1,x) \wedge$ Ethnicity-Language-Person$(z_1,w_1,x) \wedge$ Ethnicity-Language-Person$(z_1,y,w_2)$
Music-Artist-Place$(x_1,x,y) \leftarrow$ Music-Artist-Place$^3(x_1,x) \wedge$ Music-Artist-Place$(w_1,x,y)$
Music-Genre-Subgenre$(y,x,x_1) \leftarrow$ Music-Genre-Subgenre$^1(x,x_1) \wedge$ Music-Genre-Subgenre$(w_1,x,z_1) \wedge$ Music-Genre-Subgenre$(y,z_1,w_2)$
Type-Singer-Instrument$(y,x,x_2) \leftarrow$
    Type-Singer-Instrument$^1(x,x_2) \wedge$ Type-Singer-Instrument$(w_1,x,z_1) \wedge$ Type-Singer-Instrument$(y,w_2,z_1)$ |

## 7    LIMITATIONS AND FUTURE WORK

Our models currently capture only path-like relationships between constants; hence, future work could focus on learning rules with more complex structures, such as multi-branch or cyclic patterns that capture richer dependencies among multiple attributes. Furthermore, since existing datasets typically use random data splits for evaluation, we plan to develop more effective and informative benchmarks based on rule-driven fact generation. This also helps systematically investigate more diversified rule patterns by allowing controllable evaluation of rule learning performance with complex patterns. Finally, while our approach can be extended to facts with multiple missing values, a promising direction is to develop joint inference mechanisms that explicitly model correlations between missing values, to ensure consistent and semantically coherent completions.

### ETHICS STATEMENT

We have reviewed the ethics guidelines and we believe our research complies with the ICLR Code of Ethics at `https://iclr.cc/public/CodeOfEthics` in every respect. No human subjects or sensitive data were involved, and we do not foresee any ethical, privacy, or fairness concerns arising from this research.

### REPRODUCIBILITY STATEMENT

The proofs of all the lemmas and theorems are provided in Appendix C. The datasets and source codes used in our experiments are available at the anonymous GitHub repository with documentation: `https://anonymous.4open.science/r/HARL`.

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

## A  EXTENDED RELATED WORKS

**Rule Learning.**  Rule learning aims to automatically provide rules that explain the predictions of ML models. A number of approaches (Yang et al., 2017; Evans & Grefenstette, 2018; Sadeghian et al., 2019; Qu et al., 2021; Ferreira et al., 2022; Zhang et al., 2023; Wang et al., 2023b) have been proposed to extract such rules, but many of them lack formal guarantees establishing the relationship between the model and the extracted rules. Instead, they often rely on informal claims that the rules "approximate" the model's behavior (Yang et al., 2017; Evans & Grefenstette, 2018; Sadeghian et al., 2019; Qu et al., 2021).

To fill the gap, *faithful rule learning* (Tena Cucala et al., 2022a;b; 2023; Wang et al., 2024a) has been investigated to not only provide rule-based explanation, but also ensure theoretical equivalence between the model and the extracted rules. However, these methods are typically restricted to binary relational data (Tena Cucala et al., 2022b; Wang et al., 2024a), such as knowledge graphs. Extending them to tabular data is nontrivial due to the more complex relations among data cells described by each row. To the best of our knowledge, no prior work has investigated faithful rule learning for tabular data cell completion.

**Data Quality Rule Discovery.**  *Data quality rule discovery* focuses on extracting statistical dependencies that profile tables or detect inconsistencies (Chiang & Miller, 2008; Chu et al., 2014). These methods typically take a table as input and output rules describing frequent patterns, optimizing for metrics such as rule quality or confidence. Our work differs from data quality rule discovery in multiple aspects. Despite the different input, the extracted rules in our work serve as an explanatory mechanism for ML models, rather than as stand-alone quality indicators, and directly support the task of completing missing cells in multi-relational databases.

**Extending GNN to Relational Tables.**  Our task is closely related to *hypergraph link prediction* (Chen & Liu, 2023; Fatemi et al., 2020; Yadati et al., 2019; Wang et al., 2023a), since a database can be viewed as a hypergraph. Recent ML approaches extend *Graph Neural Networks (GNNs)* and propose *hypergraph neural networks* (Yadati et al., 2019; Yadati, 2020), where tabular data is represented as a hypergraph with cell values as nodes and rows as hyperedges. While effective, these methods generally lack interpretability, motivating our focus on explainable alternatives. Another line of work is *relational deep learning* (Fey et al., 2024; Chen et al., 2025), which also adapts GNNs to relational databases but they represent a database differently: rows are nodes and edges are defined by primary–foreign key relations. In contrast, both hypergraph neural networks (Yadati et al., 2019; Yadati, 2020) and our approach operate at the *cell* level, treating individual data values as nodes, which is more suitable for cell completion tasks.

**LLM-based Table Understanding.**  Recent works have explored the use of *large language models* for table understanding and reasoning (Hegselmann et al., 2023; Cheng et al., 2023; Wang et al., 2024b). However, as generative models may produce hallucinated values and cannot provide faithful explanations for their results, such uncertainty is especially problematic in safety-critical or legally regulated settings. In contrast, our approach offers provably faithful explanations for the results.

## B  FAITHFUL RULE EXTRACTION FOR THE MC MODEL WITH FIXED DATASET

Algorithm 3 outlines the process of extracting the program $\mathcal{R}_{\mathcal{M},\mathcal{D}}$ for a given MC model $\mathcal{M}$ and dataset $\mathcal{D}$. It begins with initializing $\mathcal{R}_{\mathcal{M},\mathcal{D}}$ as an empty set (line 1). $\mathcal{P}_{\text{all}}$ is initialized with all existing constants in any incomplete facts in $\mathcal{D}$ (line 2), which can also be intuitively viewed as paths of length 0. Then for length $j$ from 1 to $L$ (line 3), $\mathcal{P}_{\text{all}}$ iteratively adds expanded paths with length $j$ (lines 3–9), by considering all possible extensions for current paths with length $j-1$ (lines 7–8). The next part of the algorithm constructs a MC rule for each fact $P_h(c_{s_1}, \cdots, c_{s_{t-1}}, c_u, c_{s_{t+1}}, \cdots, c_{s_{|P_h|}})$ predicted by $\mathcal{M}$ (line 10), such that applying the rule to $\mathcal{D}$ is able to derive the same facts. The rule body is a conjunction of $\varphi_r$ with $1 \le r \le |P_h|$ and $r \ne t$ (line 11), and each $\varphi_r$ corresponds to the paths connecting $c_{s_r}$ and $c_u$. Each $\varphi_r$ is initialized as $\top$, and a function $\texttt{count}(\cdot)$ is initialized as 0 for any path schema and $\top$ (line 12). Then the algorithm counts all paths in $\mathcal{P}_{\text{all}}$ from $c_{s_r}$ to $c_u$ (line 13), grouped by the same path schema (lines 14–15). The corresponding multichain conjunction is added into $\varphi_r$ in lines 16–17. Additionally, for each $\varphi_r$, the variable $x$ is updated to match the equivalence

between $c_{s_r}$ and $c_u$ (line 18). After obtaining all $\varphi_r$ for $1 \le r \le |P_h|$ and $r \ne t$ to constitute the rule body, and updating the variables accordingly in the rule head (line 19), the rule is added to the result program (line 20).

---

**Algorithm 3:** Rule Extraction with Fixed Datasets.

---

**Input:** A MC model $\mathcal{M}$, and a dataset $\mathcal{D}$.
**Output:** A multichain program $\mathcal{R}_{\mathcal{M},\mathcal{D}}$.

1  $\mathcal{R}_{\mathcal{M},\mathcal{D}} := \emptyset$;
2  $\mathcal{P}_{\text{next}} := \{[c] \mid c \text{ appear in any incomplete fact } P_k^t(\cdots) \text{ from } \mathcal{D}\}$, $\mathcal{P}_{\text{all}} := \mathcal{P}_{\text{next}}$;
3  **foreach** $j \in \{1, \cdots, L\}$ **do**
4       $\mathcal{P}_{\text{current}} := \mathcal{P}_{next}$ , $\mathcal{P}_{\text{next}} = \emptyset$;
5       **while** $\mathcal{P}_{current}$ *is not empty* **do**
6           **pop** $[\cdots, c_{s'}]$ **from** $\mathcal{P}_{\text{current}}$;
7           **foreach** $P_k(\cdots, c_{r'_{p-1}}, c_{s'}, c_{r'_{p+1}}, \cdots) \in \mathcal{D}$ **do**
8               $\mathcal{P}_{\text{next}} = \mathcal{P}_{\text{next}} \cup \{[\cdots, c_{s'}, \mathtt{d}_{k,p \to q}, c_{r'_q}] \mid 1 \le q \le |P_k|, q \ne p\}$;
9       $\mathcal{P}_{\text{all}} := \mathcal{P}_{\text{all}} \cup \mathcal{P}_{\text{next}}$;
10 **foreach** $P_h(\cdots, c_{s_{t-1}}, c_u, c_{s_{t+1}}, \cdots)$ *completed from* $P_h^t(c_{s_1}, \cdots, c_{s_{t-1}}, c_{s_{t+1}}, \cdots, c_{s_{|P_h|}})$ **do**
11      **foreach** $r \in \{1, \cdots, t-1, t+1, \cdots, |P_h|\}$ **do**
12          $\varphi_r(x, y) := \top$ ; $\mathtt{count}(\omega) := 0$ for $\omega$ as any path schema or $\top$;
13          **foreach** $[c_{s_r}, d_1, \cdots, d_\ell, c_u] \in \mathcal{P}_{all}$ **do**
14              **foreach** $j \in \{1, \cdots, \ell\}$ **do** $(k_j, p_j, q_j) :=$ the triple corresponding to $d_j$;
15              $\omega := (k_1, p_1, q_1), \cdots, (k_\ell, p_\ell, q_\ell)$, $\mathtt{count}(\omega) := \mathtt{count}(\omega) + 1$
16          **foreach** $\omega : \mathtt{count}(\omega) > 0$ **do**
17              $\varphi_r(x, y)$ append a multichain conjunction with cardinality $\min(\mathtt{count}(\omega), \mathring{C}_\omega)$ and core being the chain pattern corresponding to $\omega$;
18          **if** $s_r \ne u$ **then** $\varphi_r := \varphi_r\{x \mapsto x_r\}$ **else** $\varphi_r := \varphi_r\{x \mapsto y\}$;
19      $H := P_h(x_1, \cdots, x_{t-1}, y, x_{t+1}, \cdots, x_{|P_h|})\{x_r \mapsto y \text{ foreach } s_r = u\}$ ,
         $A := P_h^t(x_1, \cdots, x_{t-1}, x_{t+1}, \cdots, x_{|P_h|})\{x_r \mapsto y \text{ foreach } s_r = u\}$;
20      $\mathcal{R}_{\mathcal{M},\mathcal{D}} := \mathcal{R}_{\mathcal{M},\mathcal{D}} \cup \{H \leftarrow A \wedge \bigwedge_{1 \le r \le |P_h|, r \ne t} \varphi_r\}$;
21 **return** $\mathcal{R}_{\mathcal{M},\mathcal{D}}$;

---

## C  PROOFS

**Lemma 3.4.** *Let $P_h(\mathbf{c})$ be an incomplete fact in $\mathcal{D}$ with a null value in position $1 \le t \le |P_h|$. Then, vector $\mathbf{v}_{\mathbf{c}}^{h,t}$ is equal to $\sum_{\omega \in \Omega} \mathtt{wt}_{\mathcal{M}}(\omega) \cdot \mathbf{v}_{\omega,\mathcal{D},\mathbf{c}}$, where $\mathtt{wt}_{\mathcal{M}} : \Omega \mapsto \mathbb{R}_{\ge 0}$ is a function that depends only on the parameters of $\mathcal{M}$, and $\mathbf{v}_{\omega,\mathcal{D},\mathbf{c}}$ is a non-negative vector of dimension $\epsilon$ defined as follows: (1) if $\omega = \top$, then $\mathbf{v}_{\omega,\mathcal{D},\mathbf{c}}(u)$ is the number of occurrences of $c_u$ in $\mathbf{c}$, for each $1 \le u \le \epsilon$; (2) if $\omega$ is a path schema of length $\ell \ge 1$, then $\mathbf{v}_{\omega,\mathcal{D},\mathbf{c}}(u)$ is the number of distinct paths in $\mathcal{D}$ connecting constants in $\mathbf{c}$ to $c_u$ and matching $\omega$, for each $1 \le u \le \epsilon$.*

*Proof.* Let $\mathcal{D}$ be an arbitrary incomplete database as defined in the paper that contains constants $c_1, \cdots, c_\epsilon$. Consider the computation of the MC model as Equation 1, for simplicity, we denote each adjacency matrix $\mathbf{M}_{k,p \to q}$ with $1 \le k \le \delta$, $1 \le p, q \le |P_k|$ and $p \ne q$ as $\hat{\mathbf{M}}_{\mathtt{d}_{k,p \to q}}$ where $1 \le \mathtt{d}_{k,p \to q} \le K - 1$. Besides, let $\hat{\mathbf{M}}_K \in \{0,1\}^{\epsilon \times \epsilon}$ be an identity matrix, in which $\hat{\mathbf{M}}_K(i,i) = 1$ for $1 \le i \le \epsilon$ and 0 elsewhere. Then Equation 1 can be written as Equation 4.

$$(\mathbf{v}_{c_s}^{i,j})^{\mathsf{T}} = (\mathbf{v}_{c_s}^{i,j-1})^{\mathsf{T}} \cdot \sum_{d=1}^{K} \mathbf{b}^{h,t}(i,j,d) \cdot \hat{\mathbf{M}}_d \,. \tag{4}$$

Also, the computation of the result vector $\mathbf{v}_{\mathbf{c}}^{h,t}$ can be written as Equation 5.

$$(\mathbf{v}_{\mathbf{c}}^{h,t})^{\mathsf{T}} = \sum_{i=1}^{N} \sum_{c_s \in \mathbf{c}} (\mathbf{v}_{c_s}^{i,0})^{\mathsf{T}} \cdot \prod_{j=1}^{L} \left( \sum_{d=1}^{K} \mathbf{b}^{h,t}(i,j,d) \cdot \hat{\mathbf{M}}_d \right) . \tag{5}$$

The distributive and associative properties of the sum and product operators in Equation 5 allow us to rewrite it as Equation 6.

$$(\mathbf{v}_{\mathbf{c}}^{h,t})^{\mathsf{T}} = \sum_{[d_1,\cdots,d_L]\in\{1,\cdots,K\}^L} \left(\sum_{i=1}^{N}\prod_{j=1}^{L}\mathbf{b}^{h,t}(i,j,d_j)\right) \cdot \left(\sum_{c_s\in\mathbf{c}}(\mathbf{v}_{c_s}^{i,0})^{\mathsf{T}}\right) \cdot \hat{\mathbf{M}}_{d_1} \cdot \hat{\mathbf{M}}_{d_2} \cdot \cdots \cdot \hat{\mathbf{M}}_{d_L}. \quad (6)$$

Each list $[d_1,\cdots,d_L]$ with $1 \le d_i \le K$ for $1 \le i \le L$ corresponds to a path schema or $\top$ as follows. Let $[d'_1,\cdots,d'_\ell]$ be the list of removing all occurrences of $K$s from $[d_1,\cdots,d_L]$. Then the list corresponds to $\top$ if $\ell = 0$, or a path schema if $\ell \ge 1$, where each $d'_i$ satisfying $1 \le d'_i \le K-1$ corresponds to a unique triple as $(k_i, p_i, q_i)$ by definition, and the $i$-th item of the path schema is $(P_{k_i}, p_i, q_i)$, respectively, for $1 \le i \le \ell$.

For a path schema $\omega$ with $1 \le \ell \le L$, let $S_\omega^L$ be the set of all lists $[d_1,\cdots,d_L]$ that can be obtained from $[\mathsf{d}_{k_1,p_1\to q_1},\cdots,\mathsf{d}_{k_\ell,p_\ell\to q_\ell}]$ by padding it (if needed) with the value $K$. Then Equation 6 can be written as Equation 7.

$$(\mathbf{v}_{\mathbf{c}}^{h,t})^{\mathsf{T}} = \sum_{\omega\in\Omega} \left(\sum_{[d_1,\cdots,d_L]\in S_\omega^L} \left(\sum_{i=1}^{N}\prod_{j=1}^{L}\mathbf{b}^{h,t}(i,j,d_j)\right) \cdot \left(\sum_{c_s\in\mathbf{c}}(\mathbf{v}_{c_s}^{i,0})^{\mathsf{T}}\right) \cdot \hat{\mathbf{M}}_{d_1} \cdot \hat{\mathbf{M}}_{d_2} \cdot \cdots \cdot \hat{\mathbf{M}}_{d_L}\right). \quad (7)$$

Each vector $(\mathbf{v}_{c_s}^{i,0})^{\mathsf{T}} \cdot \hat{\mathbf{M}}_{d_1} \cdot \hat{\mathbf{M}}_{d_2} \cdot \cdots \cdot \hat{\mathbf{M}}_{d_L}$ for $c_s \in \mathbf{c}$ is equal to $(\mathbf{v}_{c_s}^{i,0})^{\mathsf{T}} \cdot \hat{\mathbf{M}}_{d'_1} \cdot \cdots \cdot \hat{\mathbf{M}}_{d'_\ell}$ by removing all occurrences of $\hat{\mathbf{M}}_K$. If $\ell \ge 1$, a simple inductive argument shows that the vector describes the number of paths of length $\ell$ from $c_s$ to each constant. In particular, the $u$-th element of the vector is the number of paths with length $\ell$ from $c_s$ to $c_u$. Based on that, the $u$-th element of the vector obtained by adding up $c_s \in \mathbf{c}$ as $\left(\sum_{c_s\in\mathbf{c}}(\mathbf{v}_{c_s}^{i,0})^{\mathsf{T}}\right) \cdot \hat{\mathbf{M}}_{d_1} \cdot \hat{\mathbf{M}}_{d_2} \cdot \cdots \cdot \hat{\mathbf{M}}_{d_L}$ is the total number of paths with length $\ell$ from $c_{c_s} \in \mathbf{c}$ to each constant. If $\ell = 0$, the vector $\left(\sum_{c_s\in\mathbf{c}}(\mathbf{v}_{c_s}^{i,0})^{\mathsf{T}}\right) \cdot \hat{\mathbf{M}}_{d_1} \cdot \hat{\mathbf{M}}_{d_2} \cdot \cdots \cdot \hat{\mathbf{M}}_{d_L}$ is simply $\left(\sum_{c_s\in\mathbf{c}}(\mathbf{v}_{c_s}^{i,0})^{\mathsf{T}}\right)$, in which the $u$-th element for $1 \le u \le \epsilon$ is the number of $c_u$ that appears in $\mathbf{c}$. Moreover, for each conjunction $\omega \in \Omega$ with $\ell \ge 1$ and for each $[d_1,\cdots,d_L] \in S_\omega^L$, there is a one-to-one correspondence between each mapping of $\omega$ that grounds $x$ to each $c_s \in \mathbf{c}$ and $y$ to $c_u$, and each chain of length $\ell$ from $c_s$ to $c_u$. Therefore, for each $[d_1,\cdots,d_L] \in S_\omega^L$, the vector $\left(\sum_{c_s\in\mathbf{c}}(\mathbf{v}_{c_s}^{i,0})^{\mathsf{T}}\right) \cdot \hat{\mathbf{M}}_{d_1} \cdot \hat{\mathbf{M}}_{d_2} \cdot \cdots \cdot \hat{\mathbf{M}}_{d_L}$ is always $\mathbf{v}_{\omega,\mathcal{D},\mathbf{c}}$. Besides, for $\omega = \top$, there is a unique list $[K,\cdots,K] \in S_\omega^L$. For this list, as introduced above, the vector $\left(\sum_{c_s\in\mathbf{c}}(\mathbf{v}_{c_s}^{i,0})^{\mathsf{T}}\right) \cdot \hat{\mathbf{M}}_{d_1} \cdot \hat{\mathbf{M}}_{d_2} \cdot \cdots \cdot \hat{\mathbf{M}}_{d_L}$ is also equal to $\mathbf{v}_{\omega,\mathcal{D},\mathbf{c}}$. Therefore, in Equation 7 we can replace the vector $\left(\sum_{c_s\in\mathbf{c}}(\mathbf{v}_{c_s}^{i,0})^{\mathsf{T}}\right) \cdot \hat{\mathbf{M}}_{d_1} \cdot \hat{\mathbf{M}}_{d_2} \cdot \cdots \cdot \hat{\mathbf{M}}_{d_L}$ with $\mathbf{v}_{\omega,\mathcal{D},\mathbf{c}}$, as it is equal for each $[d_1,\cdots,d_L] \in S_\omega^L$ of a given $\omega$. Then the right side of Equation 7 becomes $\sum_{\omega\in\Omega}\mathtt{wt}_{\mathcal{M}}(\omega) \cdot \mathbf{v}_{\omega,\mathcal{D},\mathbf{c}}$, which finishes our proof with

$$\mathtt{wt}_{\mathcal{M}}(\omega) = \sum_{[d_1,\cdots,d_L]\in S_\omega^L} \left(\sum_{i=1}^{N}\prod_{j=1}^{L}\mathbf{b}^{h,t}(i,j,d_j)\right). \quad (8)$$

$\square$

**Corollary C.1.** *The equation $\mathbf{v}_{\mathbf{c}}^{h,t} = \sum_{\omega\in\Omega}\mathtt{wt}_{\mathcal{M}}(\omega) \cdot \mathbf{v}_{\omega,\mathcal{D},\mathbf{c}}$ in Lemma 3.4 can be equally written as $\mathbf{v}_{\mathbf{c}}^{h,t} = \sum_{c_s\in\mathbf{c}}\sum_{\omega\in\Omega}\mathtt{wt}_{\mathcal{M}}(\omega) \cdot \mathbf{v}_{\omega,\mathcal{D},c_s}$, where $\Omega$ and $\mathtt{wt}_{\mathcal{M}}(\omega)$ are the same as Lemma 3.4, while $\mathbf{v}_{\omega,\mathcal{D},c_s}$ is a vector that depends on $\omega$, $\mathcal{D}$, and the constant $c_s \in \mathbf{c}$.*

*Proof.* The proof of Lemma 3.4 shows that $\mathbf{v}_{\omega,\mathcal{D},\mathbf{c}} = \left(\sum_{c_s\in\mathbf{c}}(\mathbf{v}_{c_s}^{i,0})^{\mathsf{T}}\right) \cdot \hat{\mathbf{M}}_{d_1} \cdot \hat{\mathbf{M}}_{d_2} \cdot \cdots \cdot \hat{\mathbf{M}}_{d_L}$. Let $\mathbf{v}_{\omega,\mathcal{D},c_s} = (\mathbf{v}_{c_s}^{i,0})^{\mathsf{T}} \cdot \hat{\mathbf{M}}_{d_1} \cdot \hat{\mathbf{M}}_{d_2} \cdot \cdots \cdot \hat{\mathbf{M}}_{d_L}$ for each $c_s \in \mathbf{c}$. Then the distributive property of the sum and product operators allows us to exchange the order of two sum operators, which completes the proof. $\square$

**Theorem 4.4.** *The program $\mathcal{R}_{\mathcal{M}}$ extracted by Algorithm 1 is faithful to $\mathcal{M}$ when $\gamma = \beta$. Algorithm 1 terminates in $\mathcal{O}\left(C^{\delta^L}\nu^{2L}\right)$ steps with $\nu = \max_{1\le k\le\delta}|P_k|$, $C = \max_{1\le i\le|\Omega|-1}\mathring{C}_i$.*

*Proof.* Let $\mathcal{R}$ be the output of Algorithm 1 of input $\mathcal{M}$ with $\gamma = \beta$. Let $L$ and $N$ be the depth and rank of $\mathcal{M}$, respectively. To compare MC models to MC rules, we view each MC model $\mathcal{M}$ as defining a transformation $T_{\mathcal{M}}$ over datasets: given a dataset $\mathcal{D}$ representing a database, where

incomplete rows are facts over auxiliary predicates, $T_{\mathcal{M}}$ maps $\mathcal{D}$ to the set of all facts corresponding to new rows obtained by applying $\mathcal{M}$ for tabular data cell completion. We will compare $T_{\mathcal{M}}$ with the operator $T_{\mathcal{R}}$ induced by a set of rules $\mathcal{R}$. A set of rules $\mathcal{R}$ is *faithful* to $\mathcal{M}$ if $T_{\mathcal{M}}(\mathcal{D}) = T_{\mathcal{R}}(\mathcal{D})$ for each dataset $\mathcal{D}$ representing a database.

**Auxiliary Proposition 1.** We first prove an auxiliary proposition that, for each element $\omega \in \Omega$ being either a path schema or $\top$, the value $\texttt{wt}(\omega)$ computed in Algorithm 1 lines 3–8, satisfies $\texttt{wt}(\omega) = \texttt{wt}_{\mathcal{M}}(\omega)$ after the whole iteration over $[d_1, \cdots, d_L]$ with $d_i \in \{1, \cdots, K\}$. Let $\omega = (k_1, p_1, q_1) \cdots (k_\ell, p_\ell, q_\ell)$ (resp. $\omega = \top$) where $\ell \geq 1$ (resp. $\ell = 0$). Let $[d'_1, \cdots, d'_\ell] = [\texttt{d}_{k_1, p_1 \to q_1}, \cdots, \texttt{d}_{k_\ell, p_\ell \to q_\ell}]$ (resp. $[]$), where each $1 \leq \texttt{d}_{k_i, p_i \to q_i} \leq K - 1$ is the position of $(k_i, p_i, q_i)$ as defined in the paper. Note that $\texttt{wt}(\omega)$ is only updated within the iteration of $h, t$ (line 8). Meanwhile, lines 3–8 ensure that $\texttt{wt}(\omega) = \sum_{[d_1, \cdots, d_L] \in S} \left( \sum_{i=1}^{N} \prod_{j=1}^{L} \mathbf{b}^{h,t}(i, j, d_k) \right)$ where $S$ contains all lists of length $L$, satisfying that $[d_1, \cdots, d_L]$ becomes $[d'_1, \cdots, d'_\ell]$ by removing all occurrences of $K$. Therefore, when performing the iteration in lines 5–6 with the list $[d'_1, \cdots, d'_\ell]$ from $[d_1, \cdots, d_L]$, the $\omega$ is constructed in line 7, and $\texttt{wt}(\omega)$ is defined. Besides, to prove $\texttt{wt}(\omega) = \texttt{wt}_{\mathcal{M}}(\omega)$, with Equation 8 we need to show $S = S_\omega^L$, which straightforwardly follows the definitions of $S$ and $S_\omega^L$.

**Auxiliary Proposition 2.** Next, we prove another auxiliary proposition that, in each iteration with a pair of $h, t$ (lines 2–20), the set $\Theta$ (initiated in line 11) consists of all conjunctions of the form $\varphi = \phi_1 \wedge \cdots \wedge \phi_n$ and $\varphi\{x \mapsto y\} = \phi_1\{x \mapsto y\} \wedge \cdots \wedge \phi_n\{x \mapsto y\}$ with $0 \leq n \leq |\Omega| - 1$, satisfying (1) for $n \geq 1$, each $\phi_i$ for $1 \leq i \leq n$ is a multichain conjunction for $x$ and $y$ with a distinct core, (2) for $n = 0$, $\varphi = \top$ and $\varphi\{x \mapsto y\} = \top\{x \mapsto y\}$ are two distinct elements in $\Theta$. Besides, for each element $\varphi \in \Theta$, the value of $\texttt{wt}'(\varphi)$ is defined in line 15. Notice that each path schema $\Omega(i)$ for $1 \leq i \leq |\Omega| - 1$ corresponds to a distinct chain pattern. By enumerating all the lists $[C_1, \cdots, C_{|\Omega|-1}]$ with $0 \leq C_i \leq \mathring{C}_i$, line 12 produces all possible combinations $\phi_1, \cdots, \phi_{|\Omega|-1}$ with each $\phi_i$ being $\top$ or a multichain conjunction with core corresponding to $\Omega(i)$ and cardinality $C_i$ for $1 \leq i \leq |\Omega| - 1$. Then, for each combination $\phi_1, \cdots, \phi_{|\Omega|-1}$, the disjoint conjunctions $\varphi$ and $\varphi\{x \mapsto y\}$ are constructed, respectively (lines 13–14), along with the values $\texttt{wt}'(\varphi)$ and $\texttt{wt}'(\varphi\{x \mapsto y\})$ (line 15).

**Soundness.** We prove the soundness of $\mathcal{R}$ to $\mathcal{M}$ by taking an arbitrary dataset $\mathcal{D}$ that encodes a database, and showing that $T_{\mathcal{R}}(\mathcal{D}) \subseteq T_{\mathcal{M}}(\mathcal{D})$.

Let $P_h(c_{s_1}, \cdots, c_{s_{t-1}}, c_u, c_{s_{t+1}}, \cdots, c_{s_{|P_h|}})$ be an arbitrary fact in $T_{\mathcal{R}}(\mathcal{D})$. To derive the fact, there exists $P_h^t(c_{s_1}, \cdots, c_{s_{t-1}}, c_{s_{t+1}}, \cdots, c_{s_{|P_h|}}) \in \mathcal{D}$, and a MC rule $R \in \mathcal{R}$ such that the body of $R$ is of the form $A \wedge \varphi_1 \wedge \cdots \wedge \varphi_{t-1} \wedge \varphi_{t+1} \wedge \cdots \wedge \varphi_{|P_h|}$, where $A$ is an atom with predicate $P_h^t$ describing the structure of the incomplete fact, and each $\varphi_r$ for $1 \leq r \leq |P_h|$ and $r \neq t$ is either (**Case 1**) $\varphi_r = \top$, or (**Case 2**) $\varphi_r = \phi_1 \wedge \cdots \wedge \phi_P$ with $P \geq 1$ and each $\phi_p$ for $1 \leq p \leq P$ being a multichain conjunction with a distinct core; meanwhile, there exists a mapping $\sigma$ from the variables in $R$ to constants in $\mathcal{D}$ that grounds $x_r$ (resp. $y$ if $\varphi_r$ is of the form $\varphi_r\{x \mapsto y\}$) to $c_{s_r}$ and $y$ to $c_u$ such that, for each multichain conjunction $\phi_p$, if its core is a chain pattern $\Phi^p = \lambda_1^p \wedge \cdots \wedge \lambda_{\ell_p}^p$ where each $\lambda_j^p$ for $1 \leq j \leq \ell_p$ is an atom with predicate $P_{k_j}$, variables $z_{j-1}, z_j$ on the $p_j$-th, $q_j$-th positions while variables elsewhere are pairwise distinct, and the cardinality of $\phi_p$ is $C_p$, then $\lambda_j^p \sigma \in \mathcal{D}$ for each $1 \leq j \leq \ell_p$, and for each pair $j, j'$ with $1 \leq j < j' \leq C_p$, there exists $1 \leq k \leq \ell_p$ such that $z_k^j \sigma \neq z_k^{j'} \sigma$.

As each $\varphi_r$ in the body of $R$ corresponds to the constant $c_{s_r}$ for $1 \leq r \leq |P_h|$, $r \neq t$, and is independent to each other, we analyze each $\varphi_r$ by the two cases as mentioned above. Besides, by Corollary C.1, the vector $\mathbf{v}_{\mathbf{c}}^{h,t} = \sum_{1 \leq r \leq |P_h|, r \neq t} \sum_{\omega \in \Omega} \texttt{wt}_{\mathcal{M}}(\omega) \cdot \mathbf{v}_{\omega, \mathcal{D}, c_{s_r}}$ can be viewed as the sum of $|P_h| - 1$ independent parts as $\mathbf{v}_{c_{s_r}}^{h,t} = \sum_{\omega \in \Omega} \texttt{wt}_{\mathcal{M}}(\omega) \cdot \mathbf{v}_{\omega, \mathcal{D}, c_{s_r}}$, each of which corresponds to a constant $c_{s_r}$ with $1 \leq r \leq |P_h|$ and $r \neq t$. We consider the contribution of each part to the value $\mathbf{v}_{\mathbf{c}}^{h,t}(u)$ as follows.

**Case 1.** If $\varphi_r = \top$, the $\texttt{wt}'(\varphi_r)$ and $\texttt{wt}'(\varphi_r\{x \mapsto y\})$ are computed in line 15, respectively, with $[C_1, \cdots, C_{|\Omega|-1}] = [0, \cdots, 0]$. The auxiliary proposition 1 shows that $\texttt{wt}(\top) = \texttt{wt}_{\mathcal{M}}(\top)$. Therefore, if $u = s_r$, the value of $\mathbf{v}_{c_{s_r}}^{h,t}(u) = \texttt{wt}_{\mathcal{M}}(\top) = \texttt{wt}'(\varphi_r)$ and $\varphi_r$ is of the form $\varphi_r\{x \mapsto$

$y\} = \top\{x \mapsto y\}$; if $u \neq s_r$, the value of $\mathbf{v}_{c_{s_r}}^{h,t}(u) = 0 = \mathtt{wt}'(\varphi_r)$ and $\varphi_r$ is of the form $\varphi_r = \top$. Both cases satisfy that $\mathbf{v}_{c_{s_r}}^{h,t}(u) \geq \mathtt{wt}'(\varphi_r)$.

**Case 2.** If $\varphi_r$ is of the form $\phi_1 \wedge \cdots \wedge \phi_P$ with $P \geq 1$, for each multichain conjunction $\phi_p$ with $1 \leq p \leq P$, we can use $\sigma$ to produce $C_p$ mappings $\sigma_1, \cdots, \sigma_{C_p}$ defined as $\sigma_j(z_i^j) = z_i \sigma$ for each $0 \leq i \leq \ell$ and $1 \leq j \leq C_p$. If $C_p > 1$, then all mappings are necessarily pairwise distinct (i.e., they differ in the assignment of at least one variable) because line 10 ensures that $C_p > 1$ can only occur when the length of path schema has length $\geq 2$, namely, the corresponding chain pattern contains at least two body atoms. Thus, for each pair of $j$, $j'$ satisfying $1 \leq j < j' \leq C_p$, there exists $1 \leq k \leq \ell - 1$ such that $z_k^j \sigma \neq z_k^{j'} \sigma$. Therefore, these mappings provides at least $C_p$ distinct paths of the same path schema connecting $c_{s_r}$ and $c_u$. Let $\omega_p$ be the path schema corresponding to the core of $\phi_p$, then we have $\mathbf{v}_{\omega_p, \mathcal{D}, c_{s_r}}(u) \geq C_p$. By construction, $\omega_p \in \Omega$ for each $\phi_p$ with $1 \leq p \leq P$. By Corollary C.1, $\mathbf{v}_{c_{s_r}}^{h,t} = \sum_{\omega \in \Omega} \mathtt{wt}_{\mathcal{M}}(\omega) \cdot \mathbf{v}_{\omega, \mathcal{D}, c_{s_r}}$. Both $\mathtt{wt}_{\mathcal{M}}(\omega)$ and $\mathbf{v}_{\omega, \mathcal{D}, c_{s_r}}$ for all $\omega \in \Omega$ are non-negative, so $\mathbf{v}_{c_{s_r}}^{h,t}(u) \geq \sum_{1 \leq p \leq P} \mathtt{wt}_{\mathcal{M}}(\omega_p) \cdot C_p$ if $u \neq s_r$, or $\mathbf{v}_{c_{s_r}}^{h,t}(u) \geq \sum_{1 \leq p \leq P} \mathtt{wt}_{\mathcal{M}}(\omega_p) \cdot C_p + \mathtt{wt}_{\mathcal{M}}(\top)$ if $u = s_r$ (with $\varphi_r$ of the form $\varphi_r\{x \mapsto y\}$). The auxiliary proposition 1 shows that $\mathtt{wt}(\omega) = \mathtt{wt}_{\mathcal{M}}(\omega)$ for each $\omega \in \Omega$. The auxiliary proposition 2 shows that $\Theta$ consists of all candidates for $\varphi_r$ with their score $\mathtt{wt}'(\varphi_r)$. Therefore, we have $\mathbf{v}_{c_{s_r}}^{h,t}(u) \geq \mathtt{wt}'(\varphi_r)$.

Finally, as each vector $\mathbf{v}_{c_{s_r}}^{h,t}$ for $1 \leq r \leq |P_h|$ and $r \neq t$ are independent to each other, the value $\mathbf{v}_{\mathbf{c}}^{h,t}(u) = \sum_{1 \leq r \leq |P_h|, r \neq t} \mathbf{v}_{c_{s_r}}^{h,t}(u)$. Therefore, we have $\mathbf{v}_{\mathbf{c}}^{h,t}(u) \geq \sum_{1 \leq r \leq |P_h|, r \neq t} \mathtt{wt}'(\varphi_r)$. $R \in \mathcal{R}$ indicates $\sum_{1 \leq r \leq |P_h|, r \neq t} \mathtt{wt}'(\varphi_r) > \gamma$. Given $\beta = \gamma$, we have $\mathbf{v}_{\mathbf{c}}^{h,t}(u) > \beta$, indicating the fact $P_h(c_{s_1}, \cdots, c_{s_{t-1}}, c_u, c_{s_{t+1}}, \cdots, c_{s_{|P_h|}}) \in T_{\mathcal{M}}(\mathcal{D})$.

**Completeness.** We prove the completeness by taking an arbitrary dataset $\mathcal{D}$ that encodes a database, and showing that $T_{\mathcal{M}}(\mathcal{D}) \subseteq T_{\mathcal{R}}(\mathcal{D})$.

Let $P_h(c_{s_1}, \cdots, c_{s_{t-1}}, c_u, c_{s_{t+1}}, \cdots, c_{s_{|P_h|}})$ be an arbitrary fact predicted by $\mathcal{M}$ to complete the fact $P_h^t(c_{s_1}, \cdots, c_{s_{t-1}}, c_{s_{t+1}}, \cdots, c_{s_{|P_h|}}) \in \mathcal{D}$ with the constant $c_u$. We show that there exists a MC rule in $\mathcal{R}$ which derives the same fact from $\mathcal{D}$. Specifically, we consider two cases. In each case, we construct a rule of the form 2 with head $P_h(x_1, \cdots, x_{t-1}, y, x_{t+1}, \cdots, x_{|P_h|})\}\{x_r \mapsto y \text{ foreach } s_r = u\}$, and the body being the conjunction of $\varphi_r$ for $1 \leq r \leq |P_h|$ and $r \neq t$, and show that (1) this rule is in $\mathcal{R}$, and (2) this rule can be grounded in $\mathcal{D}$ with each $x_r$ (or $y$ if $s_r = u$) mapped to $c_{s_r}$ and $y$ to $c_u$. Therefore, $P_h(c_{s_1}, \cdots, c_{s_{t-1}}, c_u, c_{s_{t+1}}, \cdots, c_{s_{|P_h|}}) \in T_{\mathcal{R}}(\mathcal{D})$.

By Corollary C.1, $\mathbf{v}_{\mathbf{c}}^{h,t} = \sum_{1 \leq r \leq |P_h|, r \neq t} \sum_{\omega \in \Omega} \mathtt{wt}_{\mathcal{M}}(\omega) \cdot \mathbf{v}_{\omega, \mathcal{D}, c_{s_r}}$ can be viewed as the sum of $|P_h| - 1$ independent parts as $\mathbf{v}_{c_{s_r}}^{h,t} = \sum_{\omega \in \Omega} \mathtt{wt}_{\mathcal{M}}(\omega) \cdot \mathbf{v}_{\omega, \mathcal{D}, c_{s_r}}$. For each $\mathbf{v}_{c_{s_r}}^{h,t}$, let $\Omega_r$ be the set of all path schemas $\omega \in \Omega$ satisfying $\mathbf{v}_{\omega, \mathcal{D}, c_{s_r}}(u) > 0$, and let $\omega \in \Omega_r$ be such an arbitrary path schema. By definition, if $\omega = \top$, then $\mathbf{v}_{\omega, \mathcal{D}, c_{s_r}}(u) = 1$ if $s_r = u$ and $0$ if $s_r \neq u$; otherwise, $\mathbf{v}_{\omega, \mathcal{D}, c_{s_r}}(u)$ is the number of distinct paths of schema $\omega$ in $\mathcal{D}$ that connects $c_{s_r}$ and $c_u$. Besides, in the case of $\omega \neq \top$, let $\omega = \Omega(k)$, line 10 computes a corresponding value $\mathring{C}_k$ for it. To simplify the notations, in the following we also denote the value computed in line 10 for each $\omega$ as $\mathring{C}_\omega$.

**Case 1.** There exists $r$ and $\omega \in \Omega_r$ such that $\mathbf{v}_{\omega, \mathcal{D}, c_{s_r}}(u) > \mathring{C}_\omega$. Note that this is not the case for $\omega = \top$, because $\mathring{C}_\top$ is not defined. Also, this is not the case for $\omega$ with a single atom, because in this case $\mathbf{v}_{\omega, \mathcal{D}, c_{s_r}}(u) \leq 1$ while $\mathring{C}_\omega = 1$. Therefore, $\omega$ must have length $\geq 2$. Let $\varphi_r$ be a multichain conjunction with core being the chain pattern corresponding to $\omega$ and cardinality $\mathring{C}_\omega$. Since $\mathbf{v}_{\omega, \mathcal{D}, c_{s_r}}(u) > \mathring{C}_\omega$, there exist at least $\mathring{C}_\omega$ mappings that ground $\varphi_r$ in $\mathcal{D}$ with $x$ to $c_{s_r}$ and $y$ to $c_u$. By definition, in a multichain conjunction, each $\Phi^j$ shares no variables other than $x$ and $y$. We can take the union of those $\mathring{C}_\omega$ mappings to obtain a new mapping $\sigma$, which clearly grounds $\varphi_r$ in $\mathcal{D}$ with $x$ to $c_{s_r}$ and $y$ to $c_u$. Next, as $\omega$ is a unique element in $\Omega$, let $\omega = \Omega(k)$. Consider the list $[C_1, \cdots, C_{|\Omega|-1}]$ where $C_k = \mathring{C}_\omega$ and $C_i = 0$ for $1 \leq i \leq |\Omega| - 1$ and $i \neq k$, since this list is in the iteration of lines 12–15, we have $\varphi_r \in \Theta$ with $\mathtt{wt}'(\varphi_r)$ computed in line 15. Specifically, we have either (1) $\mathtt{wt}'(\varphi_r) = \mathtt{wt}(\omega) \cdot \mathring{C}_\omega > \beta$ (as the first part of line 15 if $u \neq s_r$), or (2) $\mathtt{wt}'(\varphi_r) = \mathtt{wt}(\omega) \cdot \mathring{C}_\omega + \mathtt{wt}(\Omega(|\Omega|)) \geq \beta$ (as the second part of line 15 if $u = s_r$). In this case,

consider the rule $P_h(x_1, \cdots, x_{t-1}, y, x_{t+1}, \cdots, x_{|P_h|}) \leftarrow P_h^t(x_1, \cdots, x_{t-1}, x_{t+1}, \cdots, x_{|P_h|}) \wedge \varphi_r$ (or replace $x_r$ both in the rule head and body with $y$ if $u = s_r$). The comparison in line 17 shows $\sum_{1 \leq i \leq |P_h|, i \neq t} \mathtt{wt}'(\varphi_i) \geq \mathtt{wt}'(\varphi_r) > \beta$. With $\beta = \gamma$ we prove the rule is in $\mathcal{R}$. Besides, we have produced a mapping as above that grounds the rule body in $\mathcal{D}$ with $x$ mapped to $c_{s_r}$ and $y$ mapped to $c_u$.

**Case 2.** For all $1 \leq r \leq |P_h|$ and $r \neq t$, there is no $\omega \in \Omega_r$ such that $\mathbf{v}_{\omega, \mathcal{D}, c_{s_r}}(u) > \mathring{C}_\omega$. For each $1 \leq r \leq |P_h|$, and $r \neq t$, let $\varphi_r$ be a conjunction of multichain conjunctions, such that for each element $\omega \in \Omega_r$ and $\omega \neq \top$, $\varphi_r$ contains a multichain conjunction with core corresponding to $\omega$, and cardinality $\mathbf{v}_{\omega, \mathcal{D}, c_{s_r}}(u)$. Such multichain conjunction is well-defined since the definition of $\Omega_r$ ensures $\mathbf{v}_{\omega, \mathcal{D}, c_{s_r}}(u) > 0$. Then we omit any $\varphi_r = \top$, and consider the MC rule whose rule body consists of remaining $\varphi_r \neq \top$ with $1 \leq r \leq |P_h|$, $r \neq t$, written as $P_h(x_1, \cdots, x_{t-1}, y, x_{t+1}, \cdots, x_{|P_h|}) \leftarrow P_h^t(x_1, \cdots, x_{t-1}, x_{t+1}, \cdots, x_{|P_h|}) \wedge \varphi_{r_1} \wedge \cdots \wedge \varphi_{r_k}\{x_r \mapsto y \text{ for each } s_r = u\}$, where $\{r_1, \cdots, r_k\} \subseteq \{1, \cdots, t-1, t+1, \cdots, |P_h|\}$.

We first consider each $\varphi_{r_i}$ with $1 \leq i \leq k$, which consists of MC conjunctions for $x_{r_i}$ and $y$ if $s_{r_i} \neq u$ or $\varphi_{r_i}\{x_{r_i} \mapsto y\}$ if $s_{r_i} = u$. For each $\omega \in \Omega_{r_i}$ and $\omega \neq \top$, there exists at least $\mathbf{v}_{\omega, \mathcal{D}, c_{s_{r_i}}}(u)$ distinct paths of schema $\omega$ in $\mathcal{D}$ that connects $c_{s_{r_i}}$ and $c_u$. By Definition 3.1, each of these paths forms a (pairwise distinct) grounding in $\mathcal{D}$ of the chain pattern corresponding to $\omega$. Note that, in a multichain conjunction by definition 4.2, each element $\Phi^j$ shares no variables other than $x$ and $y$ (i.e., $x_{r_i}$ and $y$ in $\varphi_{r_i}$). Meanwhile, each multichain conjunction in $\varphi_{r_i}$ also shares no variables other than $x_{r_i}$ and $y$. Therefore, we can take the union of the substitutions to produce a new substitution $\sigma$, which clearly grounds $\varphi_{r_i}$ in $\mathcal{D}$ that maps $x_{r_i}$ to $c_{s_{r_i}}$ and $y$ to $c_u$. Let $[C_1, \cdots, C_{|\Omega|-1}]$ be the list where $C_j = 0$ if $\Omega(j) \notin \Omega_{r_i}$, and $C_j = \mathbf{v}_{\Omega(j), \mathcal{D}, c_{s_{r_i}}}(u)$ elsewhere for $1 \leq j \leq |\Omega| - 1$. This list is well-defined since each $\omega \in \Omega_{r_i}$ is a unique element in $\Omega$. Also, this list is in the loop of line 12, as $\mathbf{v}_{\omega, \mathcal{D}, c_{s_{r_i}}}(u) \leq \mathring{C}_\omega$ for all $\omega \in \Omega_{r_i}$. We have $\varphi_{r_i}\{x_{r_i} \mapsto x\} \in \Theta$. Besides, line 15 computes $\mathtt{wt}'(\varphi_{r_i}\{x_{r_i} \mapsto x\}) = \sum_{j=1}^{|\Omega|-1} C_j \cdot \mathtt{wt}(\Omega(j))$. By Lemma 3.4 and the definition of $\Omega_{r_i}$, we have $\mathtt{wt}'(\varphi_{r_i}\{x_{r_i} \mapsto x\}) = \sum_{\omega \in \Omega_{r_i}} \mathtt{wt}_{\mathcal{M}}(\omega) \cdot \mathbf{v}_{\omega, \mathcal{D}, c_{s_{r_i}}}(u)$. Analogously, line 15 computes $\mathtt{wt}'(\varphi_{r_i}\{x_{r_i} \mapsto y\}) = \sum_{j=1}^{|\Omega|-1} C_j \cdot \mathtt{wt}(\Omega(j)) + \mathtt{wt}(\Omega(|\Omega|)) = \sum_{\omega \in \Omega_{r_i}} \mathtt{wt}_{\mathcal{M}}(\omega) \cdot \mathbf{v}_{\omega, \mathcal{D}, c_{s_{r_i}}}(u)$.

For the rule $P_h(x_1, \cdots, x_{t-1}, y, x_{t+1}, \cdots, x_{|P_h|}) \leftarrow P_h^t(x_1, \cdots, x_{t-1}, x_{t+1}, \cdots, x_{|P_h|}) \wedge \varphi_{r_1} \wedge \cdots \wedge \varphi_{r_k}\{x_r \mapsto y \text{ for each } s_r = u\}$, the analysis above shows that for each $\varphi_{r_i}$, there exists a substitution $\sigma_{r_i}$ to ground $\varphi_{r_i}$ in $\mathcal{D}$ with $x_{r_i}$ mapped to $c_{s_{r_i}}$ and $y$ to $c_u$. Definition 4.3 ensures that all $\varphi_{r_i}$ for $1 \leq i \leq k$ share no variables other than $y$. Therefore, we can take the union of these substitutions to produce a new substitution $\sigma'$, which grounds each $\varphi_{r_i}$ in $\mathcal{D}$ that maps $x_{r_i}$ to $c_{s_{r_i}}$ and $y$ to $c_u$. This shows the fact $P_h(c_{s_1}, \cdots, c_{s_{t-1}}, c_u, c_{s_{t+1}}, \cdots, c_{s_{|P_h|}})$ can be derived by the rule, given the fact $P_h^t(c_{s_1}, \cdots, c_{s_{t-1}}, c_{s_{t+1}}, \cdots, c_{s_{|P_h|}}) \in \mathcal{D}$.

To show the rule is in $\mathcal{R}$, the value compared with $\gamma$ in line 17 is $\sum_{1 \leq i \leq k} \mathtt{wt}'(\varphi_{r_i})$. With the above analysis, we have $\sum_{1 \leq i \leq k} \mathtt{wt}'(\varphi_{r_i}) = \sum_{1 \leq i \leq k} \sum_{\omega \in \Omega_{r_i}} \mathtt{wt}_{\mathcal{M}}(\omega) \cdot \mathbf{v}_{\omega, \mathcal{D}, c_{s_{r_i}}}(u)$. By the definition of $\Omega_r$ for $1 \leq r \leq |P_h|$ and $r \neq t$, this score is equal to $\sum_{1 \leq r \leq |P_h|, r \neq t} \sum_{\omega \in \Omega} \mathtt{wt}_{\mathcal{M}}(\omega) \cdot \mathbf{v}_{\omega, \mathcal{D}, c_{s_r}}(u)$, since $\mathbf{v}_{\omega, \mathcal{D}, c_{s_r}}(u) = 0$ other than $\omega \in \Omega_r$ for each $1 \leq r \leq |P_h|$, $r \neq t$. By Corollary C.1, this score is equal to $\mathbf{v}_{\mathbf{c}}^{h,t}(u)$. Since the fact $P_h(c_{s_1}, \cdots, c_{s_{t-1}}, c_u, c_{s_{t+1}}, \cdots, c_{s_{|P_h|}})$ is predicted by $\mathcal{M}$, we have $\mathbf{v}_{\mathbf{c}}^{h,t}(u) > \beta$. With $\beta = \gamma$, we have $\sum_{1 \leq i \leq k} \mathtt{wt}'(\varphi_{r_i}) > \gamma$, which shows the rule is in $\mathcal{R}$. This completes our proof.

**Time Complexity.** Let $\nu = \max_{1 \leq i \leq \delta} |P_i|$ be the maximum arity of all the predicates $P_1, \cdots, P_\delta$. Each path schema $\omega \in \Omega$ has length at most $L$, and each element of which is of the form $(P_k, p, q)$, satisfying $1 \leq k \leq \delta, 1 \leq p \leq |P_k|$ and $1 \leq q \leq |P_k|$. Therefore,

$$|\Omega| = \sum_{\ell=1}^{L} \left( \sum_{k=1}^{\delta} |P_k| \cdot (|P_k| - 1) \right)^\ell + 1 = \mathcal{O}\left( \sum_{\ell=1}^{L} \left( \delta \cdot \nu^2 \right)^\ell \right) = \mathcal{O}\left( \delta^L \cdot \nu^{2L} \right).$$

The cost to compute $\Omega$ and initialize $\mathtt{wt}(\omega)$ in line 1 is $\mathcal{O}(|\Omega| \cdot L)$. We analyze the time complexity of the main loop (lines 2–20) as follows. The number of loops over all the $h, t$ is $\sum_{k=1}^{\delta} |P_k| = \mathcal{O}(\delta \cdot \nu)$. The loop in lines 3–8 considers $K^L$ different lists. For each list, line 4 requires $\mathcal{O}(L)$

steps, and the same for lines 5–7. The computations in line 8 requires $N \cdot L$ steps. The total cost of this part is $\mathcal{O}(K^L \cdot (L + L + 1 + N \cdot L)) = \mathcal{O}(K^L \cdot N \cdot L)$. By definition of the MC model, $K = \sum_{k=1}^{\delta} |P_k|(|P_k| - 1) + 1 = \mathcal{O}(\delta \cdot \nu^2)$. So the total cost becomes $\mathcal{O}(\delta^L \cdot \nu^{2L} \cdot N \cdot L)$

The loop in lines 9-10 has $|\Omega| - 1$ iterations, each of which requires a constant number of operations. The overall cost is $\mathcal{O}(|\Omega|)$.

For the loop in lines 12–15, let $C = \max_{1 \leq i \leq |\Omega|-1} \mathring{C}_i$. The number of lists $[C_1, \cdots, C_{|\Omega|-1}]$ with $0 \leq C_i \leq \mathring{C}_i$ is bounded by $(C+1)^{|\Omega|}$, which is an upper-bound for the number of iterations. In each iteration, line 13 constructs at most $|\Omega| - 1$ multichain conjunction, where each conjunction has at most $L \cdot C$ atoms and $\binom{C}{2} \cdot L = \mathcal{O}(C^2 \cdot L)$ inequalities. Lines 14–15 combines the multichain conjunctions and computes a score with $\mathcal{O}(|\Omega|)$ operations. The total cost of this part is

$$\mathcal{O}\left((C+1)^{|\Omega|} \cdot \left(|\Omega| \cdot \left(L \cdot C + C^2 \cdot L\right) + |\Omega|\right)\right) = \mathcal{O}\left(C^{|\Omega|+2} \cdot |\Omega| \cdot L\right).$$

For the loop in lines 16–20, the size of $\Theta$ is bounded by the number of iterations as lines 12–15. In each iteration with a list $[C_1, \cdots, C_{|\Omega|-1}]$, two elements are added into $\Theta$. Therefore, we have $|\Theta| = \mathcal{O}((C+1)^{|\Omega|})$. The number of lists $[\varphi_1, \cdots, \varphi_{t-1}, \varphi_{t+1}, \cdots, \varphi_{|P_h|}]$ with each $\varphi \in \Theta$ is $|\Theta|^{|P_h|-1}$, which is bounded by $\mathcal{O}(C^{|\Omega| \cdot (\nu-1)})$. Line 17 requires at most $\nu$ operations, and the same is for lines 18–19. Line 20 requires a constant time to finish. The total cost of this part is $\mathcal{O}(C^{|\Omega| \cdot (\nu-1)} \cdot \nu)$.

Therefore, the overall time complexity of Algorithm 1 is

$$\mathcal{O}\left(|\Omega| \cdot L + \delta \cdot \nu \cdot \left(\delta^L \cdot \nu^{2L} \cdot N \cdot L + |\Omega| + C^{|\Omega|+2} \cdot |\Omega| \cdot L + C^{|\Omega| \cdot (\nu-1)} \cdot \nu\right)\right). \tag{9}$$

By replacing $|\Omega|$ with $\mathcal{O}\left(\delta^L \cdot \nu^{2L}\right)$, the expression can be simplified as

$$\mathcal{O}\left(\delta^{L+1} \cdot \nu^{2L+1} \cdot N \cdot L + L \cdot \delta^{L+1} \cdot \nu^{2L+1} \cdot C^{\delta^L \cdot \nu^{2L}+2} + \delta \cdot \nu^2 \cdot C^{\delta^L \cdot \nu^{2L} \cdot (\nu-1)}\right)$$

$$= \mathcal{O}\left(L \cdot \delta^{L+1} \cdot \nu^{2L+1} \cdot \left(N + C^{\delta^L \cdot \nu^{2L}+2}\right) + \delta \cdot \nu^2 \cdot C^{\delta^L \cdot \nu^{2L} \cdot (\nu-1)}\right) = \mathcal{O}\left(C^{\delta^L \cdot \nu^{2L}}\right). \tag{10}$$

$$\square$$

**Theorem 4.5.** *For MC model $\mathcal{M}$ and dataset $\mathcal{D}$, a program $\mathcal{R}_{\mathcal{M},\mathcal{D}} \subseteq \mathcal{R}_{\mathcal{M}}$ that returns the same output as $\mathcal{M}$ on $\mathcal{D}$ can be obtained with worst time complexity $\mathcal{O}\left(\delta^L \cdot \nu^{2L} \cdot \epsilon^{L+\nu} \cdot\right)$, for $\nu = \max_{1 \leq k \leq \delta} |P_k|$ and $\epsilon$ the number of distinct constants in $\mathcal{D}$.*

*Proof.* Let $\mathcal{R}_{\mathcal{M},\mathcal{D}}$ be the output of Algorithm 3 for the input model $\mathcal{M}$ and dataset $\mathcal{D}$. Let $L$ and $N$ be the depth and rank of $\mathcal{M}$, respectively. Besides, let $T_{\mathcal{M}}$ be the transformation defined by $\mathcal{M}$ over datasets: given a dataset $\mathcal{D}$, $T_{\mathcal{M}}$ maps $\mathcal{D}$ to the set of all facts corresponding to new rows obtained by applying $\mathcal{M}$ for completion. A set of rules $\mathcal{R}_{\mathcal{M},\mathcal{D}}$ is *faithful* to $\mathcal{M}$ if $T_{\mathcal{M}}(\mathcal{D}) = T_{\mathcal{R}_{\mathcal{M},\mathcal{D}}}(\mathcal{D})$ for each dataset $\mathcal{D}$ representing a database.

**Soundness.** We prove $T_{\mathcal{R}_{\mathcal{M},\mathcal{D}}}(\mathcal{D}) \subseteq T_{\mathcal{M}}(\mathcal{D})$ by pointing out that $\mathcal{R}_{\mathcal{M},\mathcal{D}}$ is a subset of the MC program extracted by Algorithm 1. Let $\mathcal{R}$ be the program extracted by Algorithm 1. We have $\mathcal{R}_{\mathcal{M},\mathcal{D}} \subseteq \mathcal{R}$ by construction, so $T_{\mathcal{R}_{\mathcal{M},\mathcal{D}}}(\mathcal{D}) \subseteq T_{\mathcal{R}}(\mathcal{D})$. Meanwhile, Theorem 4.4 ensures that $\mathcal{R}$ is sound for $\mathcal{M}$, i.e., $T_{\mathcal{R}}(\mathcal{D}) \subseteq T_{\mathcal{M}}(\mathcal{D})$. Therefore, we have $T_{\mathcal{R}_{\mathcal{M},\mathcal{D}}}(\mathcal{D}) \subseteq T_{\mathcal{M}}(\mathcal{D})$.

**Completeness.** To prove $T_{\mathcal{M}}(\mathcal{D}) \subseteq T_{\mathcal{R}_{\mathcal{M},\mathcal{D}}}(\mathcal{D})$, we consider an arbitrary fact $P_h(c_{s_1}, \cdots, c_{s_{t-1}}, c_u, c_{s_{t+1}}, \cdots, c_{s_{|P_h|}}) \in T_{\mathcal{M}}(\mathcal{D})$, predicted by $\mathcal{M}$ by completing the fact $P_h^t(c_{s_1}, \cdots, c_{s_{t-1}}, c_{s_{t+1}}, \cdots, c_{s_{|P_h|}}) \in \mathcal{D}$ with the constant $c_u$, and show that there exists a rule produced by Algorithm 3 which is able to derive this fact from $\mathcal{D}$. Let $H \leftarrow A \wedge \bigwedge_{1 \leq r \leq |P_h|, r \neq t} \varphi_r$ be the rule added to $\mathcal{R}_{\mathcal{M},\mathcal{D}}$ within the iteration of $P_h(c_{s_1}, \cdots, c_{s_{t-1}}, c_u, c_{s_{t+1}}, \cdots, c_{s_{|P_h|}})$ in lines 10–20. We consider each $\varphi_r$ in the rule body with $1 \leq r \leq |P_h|$ and $r \neq t$. For each multichain conjunction in $\varphi_r$ with core corresponding to path schema $\omega$, $\texttt{count}(\omega)$ counts distinct paths with schema $\omega$ from $c_{s_r}$ to $c_u$ in $\mathcal{D}$, which is greater or equal to the cardinality of this

multichain conjunction. The corresponding facts in $\mathcal{D}$ of these paths form $\mathtt{count}(\omega)$ pairwise distinct groundings of the chain pattern corresponding to $\omega$. This ensures that $\varphi_r$ can be grounded in $\mathcal{D}$ by mapping $x_r$ to $c_{s_r}$ and $y$ to $c_u$. Besides, as each $\varphi_r$ for $1 \le r \le |P_h|$ and $r \ne t$ shares no variables except $y$, each $\varphi_r$ can be grounded in $\mathcal{D}$ without affecting each other. Therefore, the whole rule can be grounded in $\mathcal{D}$ with $x_r$ mapped to $c_{s_r}$ for $1 \le r \le |P_h|$, $r \ne t$, and $y$ mapped to $c_u$. This completes our proof that $P_h(c_{s_1}, \cdots, c_{s_{t-1}}, c_u, c_{s_{t+1}}, \cdots, c_{s_{|P_h|}}) \in T_{\mathcal{R}_{\mathcal{M}},\mathcal{D}}(\mathcal{D})$.

**Time Complexity.** In the worst case, the number of possible paths in $\mathcal{P}_{\text{all}}$ reaches $\mathcal{O}(\epsilon^L \cdot K^L)$. With $K = \sum_{k=1}^{\delta} |P_k|(|P_k| - 1) + 1 = \mathcal{O}(\delta \cdot \nu^2)$, the size of $\mathcal{P}_{\text{all}}$ can be written as $\mathcal{O}(\epsilon^L \cdot \delta^L \cdot \nu^{2L})$. Consider the iteration of Algorithm 3, lines 10–20, the size of $T_{\mathcal{M}}(\mathcal{D})$ is $\mathcal{O}(\epsilon^\nu \cdot \delta)$ in the worst case (line 10), and the number of iterations over $r$ as line 11 is at most $\nu$. In lines 13–15, each loop with a specific chain $[c_{s_r}, d_1, \cdots, d_\ell, c_u]$ costs at most $\mathcal{O}(L)$ steps. The computation in lines 16–17 has at most the same cost as lines 13–15, since each specific chain $[c_{s_r}, d_1, \cdots, d_\ell, c_u]$ grounds at most one specific chain pattern in some multichain conjunction. The aggregation in lines 19–20 costs at most $\mathring{O}(L)$ steps. Besides, the computation of all the values $\mathring{C}_\omega$ for each chain pattern $\omega$ requires $\mathcal{O}(\delta^L \cdot \nu^{2L} \cdot N \cdot L)$ steps. Therefore, the worst case complexity of Algorithm 3 is

$$\mathcal{O}\left(\epsilon^L \cdot \delta^L \cdot \nu^{2L} \cdot \epsilon^\nu \cdot \delta \cdot \nu \cdot L + \delta^L \cdot \nu^{2L} \cdot N \cdot L\right) = \mathcal{O}\left(\delta^L \cdot \nu^{2L} \cdot \epsilon^{L+\nu} \cdot\right).$$

Additionally, let $T_{\mathcal{M}'}(\mathcal{D}) = \{P_h(c_{s_1}, \cdots, c_{s_{t-1}}, c_u, c_{s_{t+1}}, \cdots, c_{s_{|P_h|}})\} \subseteq T_{\mathcal{M}}(\mathcal{D})$. By replacing $T_{\mathcal{M}}(\mathcal{D})$ with $T_{\mathcal{M}'}(\mathcal{D})$ in Algorithm 3, the algorithm terminates with time complexity $\mathcal{O}(\delta^L \cdot \nu^{2L} \cdot \epsilon^L)$, and the result program $\mathcal{R}'$ contains a single rule $\rho$. The above complexity analysis for Algorithm 3 ensures that this rule can be grounded in $\mathcal{D}$ and derive $P_h(c_{s_1}, \cdots, c_{s_{t-1}}, c_u, c_{s_{t+1}}, \cdots, c_{s_{|P_h|}}) \in T_\rho(\mathcal{D})$. $\qquad\square$

**Theorem 5.2.** *The program $\mathcal{R}_{\mathcal{M}}$ extracted by Algorithm 2 is faithful to the input MC-max model $\mathcal{M}$ when $\gamma = \beta$. Algorithm 2 terminates in $\mathcal{O}\left(\delta^L \cdot \nu^{2L}\right)$ steps with $\nu = \max_{1 \le k \le \delta} |P_k|$.*

*Proof.* Let $\mathcal{R}_{\mathcal{M}}$ be the output of Algorithm 2 on input $\mathcal{M}$. Let $L$ and $N$ be the depth and rank of $\mathcal{M}$, respectively. Besides, let $T_{\mathcal{M}}$ be the transformation defined by $\mathcal{M}$ over datasets: given a dataset $\mathcal{D}$, $T_{\mathcal{M}}$ maps $\mathcal{D}$ to the set of all facts corresponding to new rows obtained by applying $\mathcal{M}$ for tabular data cell completion. A set of rules $\mathcal{R}_{\mathcal{M}}$ is *faithful* to $\mathcal{M}$ if $T_{\mathcal{M}}(\mathcal{D}) = T_{\mathcal{R}_{\mathcal{M}}}(\mathcal{D})$ for each dataset $\mathcal{D}$ representing a database.

**Soundness.** We prove the soundness of $\mathcal{R}_{\mathcal{M}}$ to $\mathcal{M}$ by taking an arbitrary input dataset $\mathcal{D}$ that encodes a database, and showing that $T_{\mathcal{R}_{\mathcal{M}}}(\mathcal{D}) \subseteq T_{\mathcal{M}}(\mathcal{D})$.

Let $P_h(c_{s_1}, \cdots, c_{s_{t-1}}, c_u, c_{s_{t+1}}, \cdots, c_{s_{|P_h|}})$ be an arbitrary fact in $T_{\mathcal{R}_{\mathcal{M}}}(\mathcal{D})$. To derive the fact, there exists a rule in $\mathcal{R}_{\mathcal{M}}$ of the form $P_h(x_1, \cdots, x_{t-1}, y, x_{t+1}, \cdots, x_{|P_h|}) \leftarrow A \wedge \lambda_{k_1,p_1 \to q_1}(x_r, z_1) \wedge \cdots \wedge \lambda_{k_\ell,p_\ell \to q_\ell}(z_{\ell-1}, y)$, or $P_h(x_1, \cdots, x_{t-1}, y, x_{t+1}, \cdots, x_{|P_h|})\{x_r \mapsto y\} \leftarrow A\{x_r \mapsto y\}$, where $1 \le r \le |P_h|$, $r \ne t$, and $A = P_h^t(x_1, \cdots, x_{t-1}, x_{t+1}, \cdots, x_{|P_h|})$. Each $\lambda_{k_i,p_i \to q_i}(z_{i-1}, z_i)$ for $1 \le i \le \ell$ denotes an atom with predicate $P_{k_i}$, variables $z_{i-1}, z_i$ on its $p_i$-th, $q_i$-th positions, and pairwise distinct variables on the other positions, respectively, with $z_0 = x_r$, $z_\ell = y$. Meanwhile, there exists a fact $P_h^t(c_{s_1}, \cdots, c_{s_{t-1}}, c_{s_{t+1}}, \cdots, c_{s_{|P_h|}}) \in \mathcal{D}$ to ground the atom $A$. Besides, in the first case where the rule body contains $\lambda_{k_1,p_1 \to q_1}(x_r, z_1) \wedge \cdots \wedge \lambda_{k_\ell,p_\ell \to q_\ell}(z_{\ell-1}, y)$ with $\ell \ge 1$, there exists a substitution $\sigma$ that ground it in $\mathcal{D}$ by mapping $x_r$ to $c_{s_r}$ and $y$ to $c_u$.

For $\omega = \lambda_{k_1,p_1 \to q_1}(x_r, z_1) \wedge \cdots \wedge \lambda_{k_\ell,p_\ell \to q_\ell}(z_{\ell-1}, y)$ with $0 \le \ell \le L$ ($\omega = \top$ for $\ell = 0$), let $S_\omega^L$ be the set of all lists $[d_1, \cdots, d_L]$ of length $L$ that can be obtained from $[\mathtt{d}_{k_1,p_1 \to q_1}, \cdots, \mathtt{d}_{k_\ell,p_\ell \to q_\ell}]$ (resp. $[]$ for $\ell = 0$) by padding it (if needed) with the value $K$. Let

$$[d_1', \cdots, d_L'] = \operatorname*{arg\,max}_{[d_1, \cdots, d_L] \in S_\omega^L} \max_{1 \le i \le N} \left(\prod_{j=1}^{L} \mathbf{b}^{h,t}(i, j, d_j)\right).$$

Since the rule is in $\mathcal{R}_{\mathcal{M}}$, we have $\prod_{j=1}^{L} \mathbf{b}^{h,t}(i, j, d_j') > \gamma$.

For simplicity, we denote each adjacency matrix $\mathbf{M}_{k,p \to q}$ with $1 \le k \le \delta$, $1 \le p, q \le |P_k|$ and $p \ne q$ as $\hat{\mathbf{M}}_{\mathtt{d}_{k,p \to q}}$ where $1 \le \mathtt{d}_{k,p \to q} \le K - 1$. Besides, let $\hat{\mathbf{M}}_K \in \{0, 1\}^{\epsilon \times \epsilon}$ be an identity matrix, in

which $\hat{\mathbf{M}}_K(i,i) = 1$ for $1 \leq i \leq \epsilon$ and 0 elsewhere. Then we can rewrite the Equation 3 and compute $\mathbf{v}_{\mathbf{c}}^{h,t}(u)$ as Equation 11.

$$
\mathbf{v}_{\mathbf{c}}^{h,t}(u) = \max_{\substack{1 \leq i \leq N \\ 1 \leq r \leq |P_h|, r \neq t}} \left( \max_{\substack{[d_1,\cdots,d_L] \in \{1,\cdots,K\}^L, \\ \hat{\mathbf{M}}_{d_1} \otimes \cdots \otimes \hat{\mathbf{M}}_{d_L}(s_r, u) = 1}} \prod_{j=1}^{L} \mathbf{b}^{h,t}(i,j,d_j) \right). \tag{11}
$$

If the rule body contains $\lambda_{k_1,p_1 \to q_1}(x_r, z_1) \wedge \cdots \wedge \lambda_{k_\ell, p_\ell \to q_\ell}(z_{\ell-1}, y)$ with $\ell \geq 1$, the existence of substitution $\sigma$ ensures that $\hat{\mathbf{M}}_{d'_1} \otimes \cdots \otimes \hat{\mathbf{M}}_{d'_L}(s_r, u) = 1$ for some $r$ with $1 \leq r \leq |P_h|$ and $r \neq t$. Otherwise, if the rule body only contains $A$, then $s_r = u$, $[d'_1, \cdots, d'_L] = [K, \cdots, K]$, and $\hat{\mathbf{M}}_{d'_1} \otimes \cdots \otimes \hat{\mathbf{M}}_{d'_L}(s_r, u) = 1$ holds as well by definition. This means the list $[d'_1, \cdots, d'_L]$ is included in the max operation of Equation 11, so we have $\mathbf{v}_{\mathbf{c}}^{h,t}(u) \geq \prod_{j=1}^{L} \mathbf{b}^{h,t}(i,j,d'_j) > \gamma$. Therefore, we have $\mathbf{v}_{\mathbf{c}}^{h,t}(u) > \beta$ with $\beta = \gamma$, which ensures $P_h(c_{s_1}, \cdots, c_{s_{t-1}}, c_u, c_{s_{t+1}}, \cdots, c_{s_{|P_h|}}) \in T_{\mathcal{M}}(\mathcal{D})$.

**Completeness.** We consider an arbitrary fact $P_h(c_{s_1}, \cdots, c_{s_{t-1}}, c_u, c_{s_{t+1}}, \cdots, c_{s_{|P_h|}}) \in T_{\mathcal{M}}(\mathcal{D})$ by completing the fact $P_h^t(c_{s_1}, \cdots, c_{s_{t-1}}, c_{s_{t+1}}, \cdots, c_{s_{|P_h|}}) \in \mathcal{D}$ with constant $c_u$, and prove the fact is in $T_{\mathcal{R}_{\mathcal{M}}}(\mathcal{D})$.

Let $[d'_1, \cdots, d'_L]$ and corresponding values of $i'$ and $r'$ be

$$
[d'_1, \cdots, d'_L], i', r' = \underset{\substack{1 \leq r \leq |P_h|, r \neq t, 1 \leq i \leq N, \\ [d_1,\cdots,d_L] \in \{1,\cdots,K\}^L, \\ \hat{\mathbf{M}}_{d_1} \otimes \cdots \otimes \hat{\mathbf{M}}_{d_L}(s_r, u) = 1}}{\arg \max} \left( \prod_{j=1}^{L} \mathbf{b}^{h,t}(i,j,d_j) \right). \tag{12}
$$

Clearly, at least one such sequence $[d'_1, \cdots, d'_L]$ with corresponding $i'$ and $r'$ must exist because $P_h(c_{s_1}, \cdots, c_{s_{t-1}}, c_u, c_{s_{t+1}}, \cdots, c_{s_{|P_h|}}) \in T_{\mathcal{M}}(\mathcal{D})$. By Equation 11, we have $\mathbf{v}_{\mathbf{c}}^{h,t}(u) = \prod_{j=1}^{L} \mathbf{b}^{h,t}(i',j,d'_j) > \beta$ and $\hat{\mathbf{M}}_{d'_1} \otimes \cdots \otimes \hat{\mathbf{M}}_{d'_L}(s_{r'}, u) = 1$. Let $[d''_1, \cdots, d''_\ell]$ be the list obtained from $[d'_1, \cdots, d'_L]$ by removing all $K$. Then we consider two cases with $\ell = 0$ and $\ell \geq 1$, respectively.

If $\ell = 0$, we consider the rule $P_h(x_1, \cdots, x_{t-1}, y, x_{t+1}, \cdots, x_{|P_h|})\{x_{r'} \mapsto y\} \leftarrow A\{x_{r'} \mapsto y\}$, where $A = P_h^t(x_1, \cdots, x_{t-1}, x_{t+1}, \cdots, x_{|P_h|})$ describes the structure of the incomplete fact. We have $\prod_{j=1}^{L} \mathbf{b}^{h,t}(i',j,d'_j) > \gamma$ with $\beta = \gamma$, which ensures the rule is in the result program $\mathcal{R}_{\mathcal{M}}$. Meanwhile, $\hat{\mathbf{M}}_{d'_1} \otimes \cdots \otimes \hat{\mathbf{M}}_{d'_L}(s_{r'}, u) = 1$ ensures that the rule can be grounded in $\mathcal{D}$ with $c_{s_{r'}} = c_u$ to derive the fact $P_h(c_{s_1}, \cdots, c_{s_{t-1}}, c_u, c_{s_{t+1}}, \cdots, c_{s_{|P_h|}}) \in T_{\mathcal{R}_{\mathcal{M}}}(\mathcal{D})$.

If $\ell \geq 1$, we consider the rule $P_h(x_1, \cdots, x_{t-1}, y, x_{t+1}, \cdots, x_{|P_h|}) \leftarrow A \wedge \lambda_{k_1,p_1 \to q_1}(x_{r'}, z_1) \wedge \cdots \wedge \lambda_{k_\ell, p_\ell \to q_\ell}(z_{\ell-1}, y)$, where $A = P_h^t(x_1, \cdots, x_{t-1}, x_{t+1}, \cdots, x_{|P_h|})$, and $(k_i, p_i, q_i)$ uniquely correspond to each index $d''_\ell$ by definition for each $1 \leq i \leq \ell$. Analogous to the case of $\ell = 0$, we have $\prod_{j=1}^{L} \mathbf{b}^{h,t}(i',j,d'_j) > \gamma = \beta$ to ensure the rule is in the result program $\mathcal{R}_{\mathcal{M}}$. Meanwhile, by definition of the adjacency matrices $\hat{\mathbf{M}}_d$ for $1 \leq d \leq K$, we have $\hat{\mathbf{M}}_{d'_1} \otimes \cdots \otimes \hat{\mathbf{M}}_{d'_L} = \hat{\mathbf{M}}_{d''_1} \otimes \cdots \otimes \hat{\mathbf{M}}_{d''_\ell}$. Therefore, we have $\hat{\mathbf{M}}_{d''_1} \otimes \cdots \otimes \hat{\mathbf{M}}_{d''_\ell}(s_{r'}, u) = 1$, which ensures that there exists a substitution to ground the rule body in $\mathcal{D}$ by mapping $x_{r'}$ to $c_{s_{r'}}$ and $y$ to $c_u$. This rule derives the fact $P_h(c_{s_1}, \cdots, c_{s_{t-1}}, c_u, c_{s_{t+1}}, \cdots, c_{s_{|P_h|}})$ and completes our proof.

**Time Complexity.** In Algorithm 2, the number of iterations over all $h, t$ with $1 \leq h \leq \delta$, $1 \leq t \leq |P_\delta|$ and rank $1 \leq i \leq N$ is $\mathcal{O}(\delta \cdot \nu \cdot N)$ with $\nu = \max_{1 \leq k \leq \delta} |P_k|$. The iteration of lines 4–7 costs $\mathcal{O}(K^L)$ steps to compute all possible scores for each chain pattern, with $K = \sum_{k=1}^{\delta} |P_k|(|P_k| - 1) + 1 = \mathcal{O}(\delta \cdot \nu^2)$. The iteration of lines 8–14 loops over all the $\mathcal{O}(K^L)$ elements of $\mathcal{S}'$, and each loop costs at most $\mathcal{O}(L + \nu)$ steps. Therefore, the time complexity of Algorithm 2 is

$$
\mathcal{O}\left(\delta \cdot \nu \cdot N \cdot K^L \cdot (L + \nu)\right) = \mathcal{O}\left(\delta^L \cdot \nu^{2L}\right).
$$

$\square$

## D    ANALYTICAL EXPRESSION OF $\mathtt{d}_{k,p\to q}$

Given predicates $P_1, \cdots, P_\delta$ with arity $|P_i| \geq 2$ for $1 \leq i \leq \delta$, the position $\mathtt{d}_{k,p\to q}$ of triple $(k, p, q)$ is computed as

$$\mathtt{d}_{k,p\to q} = \sum_{i=1}^{k-1} |P_i| \cdot (|P_i| - 1) + \begin{cases} (|P_k| - 1) \cdot (p - 1) + q & \text{if } q < p \\ (|P_k| - 1) \cdot (p - 1) + q - 1 & \text{if } q > p \end{cases}. \tag{13}$$

In turn, the triple $(k, p, q)$ can be obtained for a given $\mathtt{d}$ satisfying $1 \leq \mathtt{d} \leq K - 1$ as

$$k = \min \left\{ a \in \mathbb{Z}_{>0} \mid \sum_{i=1}^{a} |P_i| \cdot (|P_i| - 1) \geq \mathtt{d} \right\},$$

$$p = \min \left\{ b \in \mathbb{Z}_{>0} \mid \sum_{i=1}^{k-1} |P_i| \cdot (|P_i| - 1) + b \cdot (|P_k| - 1) \geq \mathtt{d} \right\}, \tag{14}$$

$$q = \mathtt{d} - \sum_{i=1}^{k-1} |P_i| \cdot (|P_i| - 1) - (p - 1) \cdot (|P_k| - 1).$$

## E    COMPARISON OF EXPRESSIVE POWER OF MC AND MC-MAX

The following proposition shows that the expressive power of MC models is not equivalent to MC-max models, as MC models can capture rules that cannot be captured by MC-max.

**Proposition E.1.** *There exists a MC model such that no MC-max model is equivalent to it, where equivalent means they produce the same result for each input dataset $\mathcal{D}$.*

*Proof.* We prove this proposition by giving an example of such MC model, and a specific input dataset $\mathcal{D}$. Then we show that no MC-max model is able to produce the same result as the MC model on this dataset.

We assume a signature consisting of a single predicate $P$ with arity $4$, and a set of constants $\{a_1, a_2, a_3, a_4, b, c, d, e, f, g, h\}$. The input dataset has an incomplete fact and three complete facts as $\mathcal{D} = \{P^4(a_1, b, c), P(a_2, b, c, d), P(a_3, b, e, f), P(a_4, g, c, h)\}$. Let the MC model $\mathcal{M} = (\mathbf{b}^{1,1}, \mathbf{b}^{1,2}, \mathbf{b}^{1,3}, \mathbf{b}^{1,4}, \beta)$ of rank $N = 1$ and depth $L = 1$, where $\beta = 0.9$, all elements of the tensors are $0$ except $\mathbf{b}^{1,4}(1, 1, \mathtt{d}_{1,2\to4}) = \mathbf{b}^{1,4}(1, 1, \mathtt{d}_{1,3\to4}) = 0.5$. Following the computation process of the MC model as Equation 1, we are able to obtain the (only) constant $d$ to replace the null value in the incomplete fact. The completed fact is $P(a_1, b, c, d)$, which can be explained by the rule 15, containing two chain patterns of length 1.

$$P(x_1, x_2, x_3, y) \leftarrow P^4(x_1, x_2, x_3) \wedge P(u, x_2, v, y) \wedge P(w, z, x_3, y). \tag{15}$$

Then we prove by contradiction that no MC-max model can produce the same result as $\mathcal{M}$ on $\mathcal{D}$. To this end, suppose there exists a MC-max model $\mathcal{M}'$ that yields the same result as $\mathcal{M}$ on $\mathcal{D}$. Then $\mathcal{M}'$ must predict the constant $d$ for the incomplete fact, since $d$ is predicted by $\mathcal{M}$. By definition of the MC-max model, it only utilizes one path with the highest weight from any existing constants (i.e., $a_1, b, c$ in this case) from the incomplete fact to the target constant. We consider the following cases.

1. The model utilizes a path connecting $a_1$ and $d$ in $\mathcal{D}$. As there is no such path in $\mathcal{D}$, this cannot be true.

2. The model utilizes a path connecting $b$ and $d$ in $\mathcal{D}$. In this case, we can find another path with the same path schema that connects $b$ and $f$. In particular, by applying the mapping $\{a_2, a_4 \mapsto a_3; g \mapsto b; c \mapsto e; d, f \mapsto h\}$ to the path, i.e., replacing each constant with the one on the same position of predicate $P$ as the fact $P(a_3, b, e, f)$. It is easy to see that the new path exists, and connects $b$ and $f$. Besides, as the path schema is unchanged, the weight of new path is the same as the one connecting $b$ and $d$. Therefore, if the model $\mathcal{M}'$ derives $d$, it must also derives $f$ in the result. This contradicts with that $\mathcal{M}$ only derives $d$ as result.

3. The model utilizes a path connecting $c$ and $d$ in $\mathcal{D}$. Analogous to case 2, we can find another path with the same path schema that connects $c$ and $h$, with the mapping $\{a_2, a_3 \mapsto a_4; b \mapsto g; e \mapsto c; d, f \mapsto h\}$. Therefore, if the model $\mathcal{M}'$ derives $d$, it must also derives $h$ in the result. This contradicts with that $\mathcal{M}$ only derives $d$ as result.

As we have enumerated all possible cases for $\mathcal{M}'$ to derive $d$, while none of them satisfies the requirement that $\mathcal{M}'$ produces the same result as $\mathcal{M}$. This contradiction completes our proof that no such MC-max model exists that can produce the same result as $\mathcal{M}$ on $\mathcal{D}$. $\qquad\square$

## F  APPLYING MC AND MC-MAX TO COMPLETE MULTIPLE NULL VALUES

In this part, we explain that our models can be applied to incomplete facts with multiple null values as a direct extension. Figure 3 shows an example of this case, where some row(s) in the input tabular data contain more than one null value to be completed. As we introduced in Section 2, input tabular data can be viewed as facts with predicates. In this example, we have predicates Name-Org.-Country, Name-City-Country-Org (abbreviated as N-O-C and N-C-C-O in the rest of this section, respectively), two complete facts N-O-C(*Emma*, *MIT*, *US*), N-C-C-O(*Alice*, *Boston*, *US*, *Amazon*), and an incomplete fact N-C-C-O(*Emma*, *Boston*, ?, ?).

| Name | Org. | Country |
|------|------|---------|
| *Emma* | *MIT* | *US* |
| ... | ... | ... |

| Name | City | Country | Org. |
|------|------|---------|------|
| *Alice* | *Boston* | *US* | *Amazon* |
| ... | ... | ... | ... |
| *Emma* | *Boston* | ? | ? |

Figure 3: An example of tabular data cell completion with multiple null values in an incomplete fact, where each '?' denotes a missing value for the data cell.

Following the same definition of data predicates $P_1, \cdots, P_\delta$ and auxiliary predicates $\{P_i^t \mid 1 \leq t \leq |P_i|\}$ introduced in Section 4, we analogously represent the input tabular data as a dataset $\mathcal{D}$. Apart from keeping the same representation for facts with zero or one null value, we extend it for facts with multiple null values. Specifically, we introduce a special constant $c_0$ that is distinct from all the constants from the input tabular data, which intuitively denotes an 'unknown' constant. Then for each incomplete fact with predicate $P_k$ and $m$ null values on its $p_1, \cdots, p_m$ positions with $1 \leq k \leq \delta$ and $1 < m < |P_k|$, $\mathcal{D}$ contains $m$ facts of the form $P_k^t(c_1, \cdots, c_{t-1}, c_{t+1}, \cdots, c_{|P_k|})$ for $t \in \{p_1, \cdots, p_m\}$, where each $c_j = c_0$ for $j \neq t$ and $j \in \{p_1, \cdots, p_m\}$. In the example of Figure 3, the incomplete fact N-C-C-O(*Emma*, *Boston*, ?, ?) is represented by two facts in $\mathcal{D}$, namely, N-C-C-O$^3$(*Emma*, *Boston*, $c_0$) and N-C-C-O$^4$(*Emma*, *Boston*, $c_0$).

As the above formulation of dataset $\mathcal{D}$ has the same form as our input introduced in Section 3, we can apply our model as Definition 3.3 to $\mathcal{D}$. In particular, by additionally encoding the constant $c_0$ as $\mathbf{v}_{c_0}^{i,0} = \{0\}^\delta$, Equation 1 is well defined. Observe that, as $c_0$ is distinct from all constants in the input tabular data, introducing $c_0$ will not affect any of the adjacency matrices $M_{k,p \to q}$. Besides, as $c_0$ is initiated as a zero vector in Equation 1, the existence of $c_0$ has neither contribution nor effect on the final results of $\mathbf{v}_c^{h,t}$.

Finally, to obtain complete facts by eliminating the existence of $c_0$, we post-process the $T_{\mathcal{M}}(\mathcal{D})$ (i.e., the result of applying model $\mathcal{M}$ to dataset $\mathcal{D}$) as follows. For each set of $m$ facts $P_k(c_1, \cdots, c_{t-1}, c_t, c_{t+1}, \cdots, c_{|P_k|})$ in $T_{\mathcal{M}}(\mathcal{D})$ with a distinct $t \in \{p_1, \cdots, p_m\}$, satisfying $c_j = c_0$ for $j \neq t$ and $j \in \{p_1, \cdots, p_m\}$, and other constants $c_i$ for $1 \leq i \leq |P_k|$ and $c_i \notin \{p_1, \cdots, p_m\}$ are the same for the $m$ facts, we merge them into a single fact without $c_0$, by substituting all $c_0$ on the $t$-th position for $t \in \{p_1, \cdots, p_m\}$ with the (only) constant $c_t \in \{c_1, \cdots, c_\delta\}$ on the same position among these $m$ facts. For example, in Figure 3, if the model produces $T_{\mathcal{M}}(\mathcal{D}) = \{$N-C-C-O(*Emma*, *Boston*, *US*, $c_0$), N-C-C-O(*Emma*, *Boston*, $c_0$, *MIT*)$\}$, the complete fact is obtained by merging the two facts as N-C-C-O(*Emma*, *Boston*, *US*, *MIT*).

To apply the MC-max model for incomplete facts with multiple null values, recall that the MC-max model has the same form and input as the MC model (see Section 5); therefore, the above extension also applies to the MC-max model. Analogously, by adding the initializing vector of constant $c_0$ as $\mathbf{v}_{c_0}^{i,0} = \{0\}^\delta$, the transformation of MC-max model as Equation 3 is well defined. Then we are able to conduct the same post-process as above to eliminate $c_0$ and obtain the complete fact.

Note that, any extension discussed in this section only involves pre-processing the input (i.e., how to get $\mathcal{D}$) and post-processing the output (i.e., how to handle $T_{\mathcal{M}}(\mathcal{D})$), while the definition of models and rule extraction algorithms are unchanged. Therefore, Theorem 4.4 and Theorem 5.2 are not affected under this extended scenario. The faithfulness between the model and extracted rules still holds.

# G EXPERIMENT SETTINGS

Additional details about the experiments are provided in this section, including the statistics of datasets, and the configurations used for training and evaluation.

## G.1 STATISTICS OF DATASETS

For relational tabular datasets, we reused the three inductive datasets WP-IND, JF-IND, and MFB-IND from Yadati (2020), as well as a transductive dataset FB-AUTO from Fatemi et al. (2020). Each dataset consists of train, validation, and test sets, where the test set of each inductive dataset contains the same predicates but also unseen constants w.r.t. the train and validation sets. Table 6 presents the statistics of each dataset. Compared with WP-IND, JF-IND, and FB-AUTO, MFB-IND has a similar number of constants but significantly more facts, indicating a much higher density than the others. In addition to the number of facts, predicates, and the arity of each predicate, we also calculated the number of occurrences for each constant in the dataset. The constants in MFB-IND exhibited a higher median frequency of occurrence than those in the other datasets. This also suggests that MFB-IND has a denser structure with more chains between pairwise constants within a given length constraint.

Table 6: Statistics of Tabular Datasets.

| | #Train Facts | #Validation Facts | #Test Facts | #Constants | #Predicates | Arity | | Constant Frequency |
| --- | --- | --- | --- | --- | --- | --- | --- | --- |
| | | | | | | median | max | (median) |
| WP-IND | 4,139 | 1,138 | 1,139 | 4,463 | 32 | 3 | 4 | 1 |
| JF-IND | 6,167 | 659 | 645 | 4,785 | 31 | 3 | 4 | 2 |
| MFB-IND | 336,733 | 15,052 | 15,056 | 3,783 | 12 | 3 | 3 | 45 |
| FB-AUTO | 6,778 | 2,255 | 2,180 | 3,388 | 8 | 5 | 5 | 4 |

To use these datasets for tabular data cell completion, each train set was randomly split into an incomplete set of facts $\mathcal{D}$ and a set of positive examples with a $3 : 1$ ratio. For inductive datasets, the test set was similarly split in an $1 : 1$ ratio. The incomplete facts with a missing value were then obtained by masking one constant at each position in the positive examples. Besides, to evaluate model performance with negative examples, for each dataset we randomly sampled negative facts by replacing the constant at each position of a positive example in the test set. Finally, an equal number of positive and negative examples were combined during the evaluation process.

For binary datasets, we reused the 12 benchmark datasets for inductive knowledge graph completion from Teru et al. (2020). Table 7 presents the statistics of each dataset. Analogous to the hypergraph datasets, we computed the number of occurrences for each constant in the dataset. Compared with NELL-995 and WN18RR, FB15k-237 datasets in general have more predicates and a higher frequency of constants, indicating a greater diversity of incomplete facts and binary connections between pairwise constants. To use these datasets for our task, we reused the splits in Wang et al. (2024a) for each train set and test set, including the incomplete set of facts, positive examples, and negative examples.

## G.2 MODEL CONFIGURATIONS

For baseline models, we reused the original implementation of HyperGCN layers[2] from Yadati et al. (2019), and adapted it to the tabular data cell completion task by adding an extra linear layer for binary classification to predict the correctness of an input fact. We used the standard version for better performance on WP-IND, JF-IND, FB-AUTO, while using the fast version on MFB-IND, as a single training epoch of the standard version on this dense dataset cost more than 24 hours.

---

[2]https://github.com/malllabiisc/HyperGCN

Table 7: Statistics of Binary datasets.

|  |  | #Train Facts | #Validation Facts | #Test Facts | #Predicates | Constant Frequency (median) |
|---|---|---|---|---|---|---|
| FB15k-237 | V1 | 4,245 | 489 | 2,198 | 180 | 3 |
|  | V2 | 9,739 | 1,166 | 4,623 | 200 | 4 |
|  | V3 | 17,986 | 2,194 | 8,271 | 215 | 5 |
|  | V4 | 27,203 | 3,352 | 13,138 | 219 | 7 |
| NELL-995 | V1 | 4,687 | 414 | 933 | 14 | 2 |
|  | V2 | 8,219 | 922 | 5,062 | 88 | 2 |
|  | V3 | 16,393 | 1,851 | 8,857 | 142 | 2 |
|  | V4 | 7,546 | 876 | 7,804 | 76 | 2 |
| WN18RR | V1 | 5,410 | 630 | 1,806 | 9 | 3 |
|  | V2 | 15,262 | 1,838 | 4,452 | 10 | 3 |
|  | V3 | 25,901 | 3,097 | 6,932 | 11 | 3 |
|  | V4 | 7,940 | 934 | 13,763 | 9 | 3 |

For GMPNN models (Yadati, 2020), we reused their original implementation[3] for our task. We reused the default hyperparameter settings for all the baseline models.

For LLM baselines, we implement Chain-of-Thought (Wei et al., 2022) with customized prompts and 3-shot examples. Each input contains an incomplete tabular data instance and a candidate fact, where the model is prompted to judge the validity of the candidate fact and output a binary answer. We accessed the GPT-5 mini model (OpenAI, 2025) via its official API. For TabLLM (Hegselmann et al., 2023), we followed the original implementation[4] and adapted it to our task. We further replaced its backbone LLM with a more advanced open-sourced model Llama-3.1-8B-Instruct (Meta, 2024). We accessed the model from HuggingFace[5] and deployed it locally for fine-tuning and evaluation. The model was fine-tuned with QLora (Dettmers et al., 2023) for binary classification of the input candidate fact.

We implemented the MC and MC-max models based on Python 3.8[6] and PyTorch 2.0[7]. For training the MC and MC-max models, we applied the Adam optimizer with cross-entropy loss, and tuned the learning rate between 0.01 and 0.0001 for each model. Each model was trained for up to 10 epochs, and an early stopping strategy was employed when the validation loss increased during the training process.

For MC and MC-max, we set depth $L = 2$ and evaluated ranks $1 \leq N \leq 3$. The threshold $\beta \in (0, 1)$ was tuned for each model to maximize the F1 score on validation sets. On binary datasets, we compared their best performance by tuning rank $N$ between 1 and 3.

For rule extraction, we implemented Algorithm 3 for the MC model on all datasets used in our experiments. We also implemented Algorithm 2 for the MC-max models and set the rule extraction threshold $\gamma = \beta$.

All the experiments were performed on a workstation with an Intel Xeon E5-2670 CPU and a Quadro RTX 8000 GPU.

## H  ADDITIONAL EVALUATION RESULTS

We provide more evaluation results in this section, including the training time of each model on every dataset, and rule extraction time of MC and MC-max models.

### H.1  EXTENDED RESULT ANALYSIS FOR TABULAR DATA CELL COMPLETION

**Sum vs. Max Aggregation.**  Results in Table 1 and Table 2 reflect the different behaviors of the two models. The MC model, with sum aggregation, often achieves higher recall by leveraging all

---

[3]`https://github.com/naganandy/G-MPNN-R`
[4]`https://github.com/clinicalml/TabLLM/tree/main`
[5]`https://huggingface.co/meta-llama/Llama-3.1-8B-Instruct`
[6]`https://www.python.org/downloads/release/python-3817/`
[7]`https://pytorch.org/get-started/pytorch-2.0/`

evidence paths, which is beneficial when multiple weak signals contribute to a correct prediction, but may be more susceptible to noise from irrelevant paths. In contrast, the MC-max model, with max aggregation, considers only the strongest path, making it more robust in noisy settings and yielding more precise and concise rules, at the cost of reduced expressivity.

**Few-shot Prompting vs. Fine-tuning LLMs.** Table 1 also demonstrates the unbalanced and unstable performance of LLM-based methods. In particular, predictions by few-shot prompting models are often overly conservative, leading to frequent failures in identifying the correct constant to fill the missing cell. In contrast, fine-tuned models tend to be overly optimistic, producing a large number of false positives. The results highlight the difficulty of calibrating LLM predictions for tabular data cell completion, where both under- and over-prediction significantly undermine reliability.

In our task, the candidate space is large and evidence is distributed across multiple tables or hops. Therefore, correct completions often require aggregating several weak, schema-constrained clues. With fixed few-shot prompting, the LLM's probability distribution is spread over many candidates and it struggles to bind constants across hops, so many true positives are filtered out, yielding conservative behavior and frequent failures to identify the correct constant. In contrast, fine-tuning makes the model internalize spurious co-occurrences and frequency priors (e.g., frequent values in the table) and loses its implicit "abstain" behavior, leading to overconfident predictions and inflated false positives. We observe that few-shot models often fail when correct predictions require multi-hop joins, while fine-tuned models frequently produce false positives by predicting overly common values for the missing cell. This contrasts with our approach that explicitly aggregates paths under schema constraints, thus improving calibration.

Compared with LLM-based methods, our models not only achieve a more balanced overall performance across all metrics, but also provide faithful, human-understandable explanations.

## H.2 TRAINING TIME

Table 8 reports the training time for each model on the WP-IND, JF-IND, MFB-IND, and FB-AUTO datasets. Generally, all models were able to finish training within several minutes to a few hours, varying among datasets. The HyperGCN model spent less training time than the GMPNN models, despite its less satisfying performance for our task. The GMPNN-sum, GMPNN-mean, and GMPNN-max models used similar training time, since their only difference is the strategy for aggregating neighborhood information.

For the MC and MC-max models, the number of learnable parameters grows linearly with the rank, resulting in the longer training time as the rank $N$ increases. This indicates the model's flexibility to scale to larger datasets by adjusting the rank as a hyperparameter. The MC-max model spent longer training time than the MC model with the same rank, primarily due to the computation of max-product and max aggregation. Additionally, MC models were observed to early stop more frequently than MC-max models with increments of validation loss. Overall, both MC and MC-max models demonstrated comparable training time to the baseline models, suggesting the practical feasibility of our approaches.

## H.3 SIGNIFICANT TEST

Based on the results in Table 1, we conducted paired t-tests for each pair of baseline and our proposed models across all evaluation metrics where MC and MC-max outperformed the baselines. A paired t-test typically indicates a statistically significant difference when $p < 0.05$. Table 9 reports the results of $p$-values, with all values of $p < 0.05$ marked with an asterisk (*). Overall, the results reveal significant differences between our proposed models, MC and MC-max, and most baseline models. In particular, both MC and MC-max significantly outperform all non-LLM baseline models in terms of precision and accuracy, and also significantly outperform GMPNN-sum in terms of F1 score. Additionally, MC-max significantly exceeds most baselines except GMPNN-max in terms of precision, recall, and AUC.

Table 8: Training Time (minutes) for Each Model.

|  | WP-IND | JF-IND | MFB-IND | FB-AUTO |
|---|---|---|---|---|
| HyperGCN | 9.3 | 18.4 | 225.0 | 39.7 |
| GMPNN-sum | 21.7 | 24.5 | 439.9 | 31.6 |
| GMPNN-mean | 21.6 | 24.4 | 438.0 | 79.0 |
| GMPNN-max | 21.3 | 24.2 | 432.1 | 77.7 |
| TabLLM (Llama 3.1) | 95.6 | 110.3 | 150.8 | 100.2 |
| MC ($L = 2, N = 1$) | 9.0 | 7.5 | 131.6 | 4.0 |
| MC ($L = 2, N = 2$) | 19.7 | 15.7 | 307.5 | 6.2 |
| MC ($L = 2, N = 3$) | 29.4 | 24.4 | 408.1 | 12.0 |
| MC-max ($L = 2, N = 1$) | 27.1 | 38.6 | 301.2 | 16.1 |
| MC-max ($L = 2, N = 2$) | 52.4 | 76.2 | 620.7 | 33.3 |
| MC-max ($L = 2, N = 3$) | 75.0 | 112.9 | 828.0 | 50.3 |
| MC-max ($L = 3, N = 1$) | 46.2 | 51.7 | 540.3 | 30.6 |
| MC-max ($L = 3, N = 2$) | 87.8 | 92.2 | 1,021.6 | 61.3 |
| MC-max ($L = 3, N = 3$) | 124.4 | 154.9 | 1,386.3 | 89.1 |

Table 9: $p$-values of Paired t-Test for Results in Table 1.

| ($L = 2$ for MC and MC-max) | | MC ($N = 1$) | MC ($N = 2$) | MC ($N = 3$) | MC-max ($N = 1$) | MC-max ($N = 2$) | MC-max ($N = 3$) |
|---|---|---|---|---|---|---|---|
| HyperGCN | P | 0.029* | 0.033* | 0.045* | 0.002* | 0.002* | 0.002* |
|  | Acc | 0.046* | 0.050 | 0.055 | 0.017* | 0.017* | 0.021* |
|  | AUC | 0.070 | 0.079 | 0.076 | 0.034* | 0.033* | 0.039* |
|  | F1 | 0.063 | 0.067 | 0.063 | 0.051 | 0.049* | 0.056 |
| GMPNN-sum | P | 0.017* | 0.019* | 0.028* | 0.007* | 0.008* | 0.006* |
|  | Acc | 0.016* | 0.016* | 0.023* | 0.005* | 0.005* | 0.004* |
|  | AUC | 0.016* | 0.016* | 0.017* | 0.002* | 0.001* | 0.001* |
|  | F1 | 0.038* | 0.036* | 0.042* | 0.020* | 0.022* | 0.021* |
| GMPNN-mean | P | 0.018* | 0.019* | 0.029* | 0.006* | 0.007* | 0.005* |
|  | Acc | 0.021* | 0.021* | 0.031* | 0.004* | 0.005* | 0.003* |
|  | AUC | 0.166 | 0.202 | 0.192 | 0.029* | 0.024* | 0.031* |
|  | F1 | 0.576 | 0.598 | 0.579 | 0.386 | 0.366 | 0.402 |
| GMPNN-max | P | 0.036* | 0.040* | 0.055 | 0.029* | 0.029* | 0.024* |
|  | Acc | 0.050 | 0.056 | 0.070 | 0.024* | 0.021* | 0.022* |
|  | AUC | 0.414 | 0.486 | 0.459 | 0.083 | 0.074 | 0.061 |
|  | F1 | 0.640 | 0.696 | 0.659 | 0.280 | 0.261 | 0.301 |
| CoT (GPT-5 mini, 3 shot) | P | 0.442 | 0.425 | 0.480 | 0.747 | 0.870 | 0.745 |
|  | R | 0.060 | 0.058 | 0.058 | 0.075 | 0.078 | 0.076 |
|  | Acc | 0.200 | 0.204 | 0.215 | 0.112 | 0.112 | 0.116 |
|  | F1 | 0.103 | 0.102 | 0.104 | 0.088 | 0.089 | 0.090 |
| CoT (GPT-5 mini, 5 shot) | P | 0.384 | 0.369 | 0.423 | 0.613 | 0.733 | 0.615 |
|  | R | 0.065 | 0.063 | 0.063 | 0.081 | 0.084 | 0.082 |
|  | Acc | 0.217 | 0.223 | 0.235 | 0.123 | 0.122 | 0.128 |
|  | F1 | 0.112 | 0.111 | 0.113 | 0.095 | 0.096 | 0.097 |
| TabLLM (Llama 3.1) | P | 0.004* | 0.004* | 0.005* | 0.000* | 0.000* | 0.000* |
|  | R | 0.072 | 0.065 | 0.073 | 0.125 | 0.144 | 0.108 |
|  | Acc | 0.010* | 0.010* | 0.012* | 0.004* | 0.005* | 0.005* |
|  | F1 | 0.029* | 0.029* | 0.029* | 0.023* | 0.024* | 0.025* |

## H.4 RULE EXTRACTION TIME FOR MC MODEL OVER A SPECIFIC DATASET

Table 10 presents the runtime of Algorithm 3 for MC models with each input tabular dataset. Since the number of results obtained by applying $\mathcal{M}$ to $\mathcal{D}$ varies across datasets, we measured the average rule extraction time per 100 result facts obtained by applying $\mathcal{M}$ to $\mathcal{D}$, to enable a consistent comparison across datasets.

In all cases, Algorithm 3 completed rule extraction within a few minutes. Besides, increasing the rank $N$ did not significantly affect the time for rule extraction. These results confirm the practical feasibility of our rule extraction approach. Additionally, we observed that the majority of the runtime was spent on computing the weights of every chain pattern and their cardinality upper bounds, while the time increment would remain limited with growing size of the result set by applying $\mathcal{M}$ to $\mathcal{D}$.

Table 10: Rule Extraction Time of MC on Fixed Datasets (seconds per 100 result facts).

|  |  | $N = 1$ | $N = 2$ | $N = 3$ |
|---|---|---|---|---|
| WP-IND |  | 11.89 | 14.61 | 15.70 |
| JF-IND |  | 11.34 | 13.78 | 14.48 |
| MFB-IND |  | 5.66 | 5.95 | 6.27 |
| FB-AUTO |  | 0.61 | 0.70 | 0.78 |
| FB15k-237 | V1 | 95.80 | 105.02 | 117.08 |
|  | V2 | 134.17 | 149.70 | 169.26 |
|  | V3 | 174.61 | 196.80 | 227.41 |
|  | V4 | 188.42 | 217.78 | 246.52 |
| NELL-995 | V1 | 0.17 | 0.20 | 0.24 |
|  | V2 | 10.71 | 12.59 | 14.30 |
|  | V3 | 47.18 | 57.08 | 64.25 |
|  | V4 | 6.97 | 8.45 | 9.42 |
| WN18RR | V1 | 0.03 | 0.05 | 0.06 |
|  | V2 | 0.05 | 0.07 | 0.10 |
|  | V3 | 0.08 | 0.12 | 0.20 |
|  | V4 | 0.05 | 0.08 | 0.16 |

## H.5 EXTENDED ANALYSIS OF EXTRACTED RULES

Table 11 presents representative rules extracted from the MC-max model with chain patterns of length 3 under $L = 3$ and $N = 3$. While longer rule bodies exhibiting more complex dependencies, some of them appear semantically redundant, and can be entailed by combinations of shorter rules. For example, the first rule in Table 11 can be interpreted as: "a type of music that belongs to a subgenre of some subgenre of a kind must also belong to the genre two levels higher." This can be entailed by a shorter rule that captures the transitive nature of the subgenre relation, together with the incomplete fact that a piece of music belongs to a particular genre.

Moreover, since our models can capture rules with chain lengths strictly shorter than $L$, we observe that many top-ranked rules are indeed shorter than the maximum allowed length. For example, in the MFB-IND dataset, $49.7\%$ of the rules extracted under $L = 3$ have actual lengths smaller than 3. This observation is consistent with the results in Table 1, where the MC-max model with $L = 3$ did not outperform the ones with $L = 2$ on MFB-IND. In practice, users may adjust $L$ based on the observed distribution of path lengths in their database to balance model expressivity and rule conciseness.

Table 11: Rules Learned by MC-max from Tabular Datasets with $L = 3$.

```
Music-Genre-Subgenre(y,x,x₁) ← Music-Genre-Subgenre¹(x,x₁)
    ∧Music-Genre-Subgenre(w₁,x,z₁) ∧ Music-Genre-Subgenre(w₂,z₁,z₂) ∧ Music-Genre-Subgenre(y,w₃,z₂)
Music-Genre-Subgenre(x,x₁,y) ← Music-Genre-Subgenre³(x,x₁)
    ∧Music-Genre-Subgenre(w₁,x,z₁) ∧ Music-Genre-Subgenre(w₂,z₁,z₂) ∧ Music-Genre-Subgenre(w₃,z₂,y)
Person-Relative-Kinship(x,x₁,y) ← Person-Relative-Kinship³(x,x₁)
    ∧Person-Relative-Kinship(x,w₁,z₁) ∧ Person-Relative-Kinship(z₁,w₂,z₂) ∧ Person-Relative-Kinship(z₂,w₃,y)
Politician-Position-Predecessor-Successor(x,x₁,y,x₂) ← Politician-Position-Predecessor-Successor³(x,x₁,x₂)
    ∧Politician-Position-Predecessor-Successor(x,w₁,w₂,z₁) ∧ Politician-Position-Predecessor-Successor(z₂,w₃,w₄,z₁)
    ∧Politician-Position-Predecessor-Successor(y,w₅,w₆,z₂)
```

Table 12 further presents top-ranked rules learned by MC-max from binary datasets. These rules are generally intuitive and easy for human users to interpret. They reflect meaningful relational schemas in the underlying knowledge graphs, such as the reflexive relation of `spouse`, transitive pattern of `also-see`, and subsumption relation of `subpart-of` and `location-located-within-location`.

Table 12: Rules Extracted from MC-max on Binary Datasets.

| Binary Datasets | |
| --- | --- |
| FB15k-237 | $\text{administrative-parent}(y,x) \leftarrow \text{contains}(x,y)$ 
 $\text{director-film}(y,x) \leftarrow \text{film-edited-by}(x,y)$ 
 $\text{language-spoken-in-country}(y,x) \leftarrow \text{official-language}(x,y)$ 
 $\text{nominated-for-award}(x,y) \leftarrow \text{award-winner}(y,x)$ 
 $\text{spouse}(y,x) \leftarrow \text{spouse}(x,y)$ |
| NELL-995 | $\text{company-economic-sector}(y,x) \leftarrow \text{subpart-of}(x,y)$ 
 $\text{organization-hired-person}(y,x) \leftarrow \text{works-for}(x,y)$ 
 $\text{subpart-of}(x,y) \leftarrow \text{location-located-within-location}(x,y)$ 
 $\text{synonym-for}(y,x) \leftarrow \text{organization-also-known-as}(x,y)$ 
 $\text{team-plays-against-team}(y,x) \leftarrow \text{team-plays-against-team}(x,y)$ |
| WN18RR | $\text{also-see}(x,y) \leftarrow \text{also-see}(x,z) \wedge \text{also-see}(z,y)$ 
 $\text{derivationally-related-form}(x,y) \leftarrow \text{derivationally-related-form}(y,x)$ 
 $\text{derivationally-related-form}(y,x) \leftarrow \text{verb-group}(x,z) \wedge \text{verb-group}(z,y)$ 
 $\text{has-part}(x,y) \leftarrow \text{hypernym}(z,x) \wedge \text{hypernym}(y,z)$ 
 $\text{verb-group}(y,x) \leftarrow \text{verb-group}(x,y)$ |

