# OpenReview forum: "Faithful Rule Learning for Tabular Data Cell Completion"
_ICLR.cc/2026/Conference — Submitted to ICLR 2026_

### Official Review · Reviewer_kFGd · 2025-10-21

**Soundness:** 4
**Presentation:** 3
**Contribution:** 4
**Rating:** 8
**Confidence:** 2

**Summary:**

This paper introduces two novel, interpretable machine learning models, MC (Multi-Chain) and MC-max, for the task of tabular data cell completion, which aims to infer missing values in table rows. Addressing the limited interpretability of existing ML models , this work proposes models that predict missing constants by learning Datalog rules describing chain-like relational patterns. The two models utilize different aggregation strategies—sum aggregation (MC) and max aggregation (MC-max)—offering distinct trade-offs between expressive power and the complexity of rule extraction. The primary contribution is that both models are fully interpretable and include faithful rule extraction algorithms. These algorithms provide formal guarantees that the extracted set of Datalog rules is equivalent to the model, ensuring that the rules produce the exact same output as the model for any input. Evaluations on standard benchmarks show that the models achieve state-of-the-art performance while offering superior interpretability.

**Strengths:**

1. Strong Formal Guarantees: The paper's most salient strength is its formal commitment to faithfulness. The authors do not merely claim interpretability; they formally prove via Theorems 4.4 and 5.2 that the extracted Datalog rule sets produce the exact same output as the models themselves (MC and MC-max) on any given input.
2. Clear Model Design and Trade-offs: The paper designs two models with clear semantics. The MC model uses sum aggregation, corresponding logically to counting the number of matching paths. The MC-max model uses max aggregation, corresponding to detecting the existence of the strongest path.
3. Excellent Empirical Performance: The experimental section is convincing. The MC and MC-max models achieve SOTA results on multiple key metrics across several tabular and binary relation datasets. They not only surpass specialized GNN baselines but also significantly outperform strong LLM baselines. This demonstrates that the models achieve interpretability without sacrificing predictive performance.

**Weaknesses:**

1. Scalability of Global Faithfulness for MC Model: Despite the formal guarantee (Theorem 4.4), the global rule extraction algorithm for the MC model (Algorithm 1) has a time complexity of $\mathcal{O}(C^{\delta^{L}\nu^{2L}})$. This complexity is doubly exponential in both the path length $L$ and the maximum table arity $\nu$. This implies that for database schemas of moderate complexity (e.g., slightly more tables or longer dependency chains), extracting the complete, equivalent Datalog program is computationally infeasible. Although database-specific extraction (Algorithm 3) is feasible, this limitation diminishes the practical significance of the MC model's "global faithfulness".
2. Rigid Dependence of Model Architecture on Database Schema: The dimension $K$ of the model's parameter tensor $b^{h,t}$ depends on all predicates (tables) and their arities (column counts) in the database. This means the model architecture is hard-bound to a specific database schema. If a new table (new predicate) is added to the database, or if the arity of an existing table changes, the entire model must be redesigned and retrained. This limits the model's applicability in dynamic database environments.

**Questions:**

Handling of Multiple Missing Values (Appendix F): The method in Appendix F for handling multiple missing values involves creating a separate auxiliary fact for each missing value and filling other missing positions with a special constant $c_0$. This approach seems to implicitly assume that the missing values are mutually independent. If the missing values are highly correlated (e.g., "City" and "Country" are missing simultaneously), could this independent prediction method lead to inconsistent completion results (e.g., completing as "Paris" and "Germany")?

---

> ### Author Response · Authors · 2025-11-20
> **Response for Reviewer kFGd**
>
> We thank the reviewer for the valuable feedback.
>
> **Weakness 1**
> > Scalability of global faithfulness for MC models.
>
> **Response.**
>
> While global rule extraction for the MC model (Algorithm 1) can be computationally expensive, our experiments show that dataset-specific extraction is efficient in practice (Table 10). Moreover, global extraction for MC-max is computationally feasible (Table 3).
>
> **Weakness 2**
> > The model architecture is hard-bound to a specific database schema.
>
> **Response.**
>
> Yes. While our models are independent of the input data instances, the architecture is schema-specific (a standard assumption in rule-learning and rule-extraction work, e.g., [1], [2]). In practice, schemas change far less frequently than data, and training MC or MC-max from scratch takes only a few hours on a single CPU (Table 8), making retraining feasible in response to schema changes. Moreover, as Definition 3.3 shows, parameters are separated by predicate–position
> pairs, suggesting the potential for incremental updates when new predicates are
> added (an interesting direction for future work).
>
> [1] D. Tena Cucala et al. Faithful Approaches to Rule Learning. KR 2022.
>
> [2] D. Tena Cucala et al. Explainable GNN-Based Models over Knowledge Graphs. ICLR 2022.
>
> **Question**
> > The method in Appendix F for handling multiple missing values involves creating a separate auxiliary fact for each missing value and filling other missing positions with a special constant. This approach seems to implicitly assume that the missing values are mutually independent.
>
> **Response.**
>
> Our approach indeed assumes independence among missing values.
> Extending the framework to model correlated missing attributes is an interesting direction for future work.

---

> > ### Comment · Reviewer_kFGd · 2025-11-21
> >
> > Thank you for the response, which has addressed my concerns. I have raised my confidence score from 2 to 3.

---

### Official Review · Reviewer_dxt1 · 2025-10-28

**Soundness:** 2
**Presentation:** 1
**Contribution:** 2
**Rating:** 4
**Confidence:** 2

**Summary:**

This paper introduces two models (MC and MC-max) for tabular data cell completion, with formal guarantees of faithfulness between model predictions and extracted Datalog rules. The paper focus on the challenge of completing missing cells in tables while providing transparent, human-readable explanations. The paper includes evaluations on several benchmarks

**Strengths:**

1) The proposed MC and MC-max models introduce reasonable approaches to capturing relational patterns in tabular data.

2) The paper evaluates on multiple standard benchmarks.

3) The paper includes detailed mathematical formulations, algorithm descriptions, and proofs

**Weaknesses:**

1) The paper suffers from significant readability issues. The mathematical notation is excessively dense, with nested subscripts and complex symbolic expressions that make the core ideas difficult to grasp. Key concepts like path schemas, multi-chain conjunctions, and the faithful extraction process are explained using notation-heavy formulations that obscure the intuitive understanding.

2) While the theoretical proofs are substantial, the paper provides minimal discussion of practical application scenarios. There's insufficient explanation of how these methods would be deployed in real-world data completion tasks, what types of tables benefit most from this approach, or practical limitations.

3)  The paper overlooks important recent developments in table understanding, particularly methods like Chain-of-Table, and other LLM-based table reasoning approaches that have shown strong performance on similar tasks. This omission limits the contextual positioning of the contributions.

**Questions:**

How about the efficiency Issues in rule extraction? Would the time cost be very high?

More intuitive explanations and visual examples would help to follow the paper.

---

> ### Author Response · Authors · 2025-11-20
> **Response for Reviewer dxt1**
>
> We thank the reviewer for the valuable comments.
>
> **Weakness 1**
> > The mathematical notation is dense. Key concepts are explained using notation-heavy formulations that obscure the intuitive understanding.
>
> **Response.**
>
> To improve readability, we have included additional examples for each key concept (paths, path schemas, chain patterns, multichain conjunctions) based on the database in Figure 2(a). We hope these examples will help clarify the intuition behind each formal component without sacrificing rigor. In Section 4.2 and Section 5, we also illustrate the general process of rule extraction for MC and MC-max models.
>
> **Weakness 2**
> > Discussion of practical application scenarios.
>
> **Response.**
>
> Because of the formal faithfulness guarantee, our approach is well suited to trust-critical data completion settings—such as knowledge curation, medical or financial record completion, and relational database repair, where interpretability is essential. As shown in Section 6 and Appendix H, both MC and MC-max also offer practical runtime and strong predictive performance across diverse datasets, supporting real-world deployment.
> We have expanded the introduction to highlight such application scenarios.
>
>
> **Weakness 3**
> > More LLM-based methods for table understanding, such as Chain-of-Table.
>
> **Response.**
>
> Our approach provides *formal guarantee of faithfulness*, ensuring that the extracted rules exactly reproduce model predictions. In contrast, LLM-based methods lack such formal guarantee and may generate inconsistent or unverifiable results.
> Despite these fundamental differences, we have included representative LLM baselines for performance comparison. The results suggest that, in addition to providing formal guarantees, our models also achieve competitive performance.
>
> The referenced Chain-of-Table work targets question answering and natural language fact verification, which differ from the tabular cell completion task we study.
>
> **Question**
> > How about the efficiency issues in rule extraction? Would the time cost be very high?
>
> **Response.**
>
> Rule extraction times for MC-max and MC models are reported in Table 3 (Section 6) and Table 10 (Appendix H), respectively. In all cases, rule extraction completed within a few minutes. These results validate the feasibility of our approaches in practice.

---

### Official Review · Reviewer_ez8B · 2025-10-31

**Soundness:** 3
**Presentation:** 2
**Contribution:** 3
**Rating:** 6
**Confidence:** 2

**Summary:**

This paper introduces interpretable machine learning models for predicting missing cells in tabular data. The authors propose two rule-based frameworks: Multi-Chain (MC) and MC-max, that infer missing values by aggregating paths among table entries, learning Datalog rules that formally mirror model behavior. Both models are fully interpretable and come with theoretical guarantees of faithfulness, ensuring that each prediction can be exactly derived from the extracted rules. MC uses sum aggregation for expressive power, while MC-max employs max aggregation for efficiency and simpler rule extraction. Extensive experiments on multiple benchmark datasets demonstrate that these models achieve state-of-the-art performance while offering transparent, human-readable explanations of their predictions.

**Strengths:**

1. A theoretically grounded and practically promising approach is proposed to interpretable rule learning for tabular data, which is a valuable contribution.
2. Experiments on multiple benchmarks are propsed to demonstrate the advantages of the prposed method.

**Weaknesses:**

1. This paper is written in a way that is somewhat difficult to follow, which makes it particularly unfriendly for readers who are not specialists in this area.
2. Comparison with recent LLM-based reasoning models could be expanded beyond few-shot prompting (e.g., reasoning-tuned LLMs or neuro-symbolic hybrids).
3. The paper provides rule examples but no quantitative results; including such results could enhance the paper’s readability and help readers better understand the approach.
4. I am not sure whether this is a major issue, but the font used in the paper appears to differ from that of the official ICLR template.

**Questions:**

Please see the Weaknesses section.

---

> ### Author Response · Authors · 2025-11-20
> **Response for Reviewer ez8B**
>
> We thank the reviewer for their comments and for pointing out the minor issue with font type, which we have fixed in the updated version.
>
> **Weakness 1**
> > This paper is somewhat difficult to follow.
>
> **Response.**
>
> To improve readability, we have included additional examples for each key definition (paths, path schemas, chain patterns, multichain conjunctions) based on the database in Figure 2(a). We hope these examples will help clarify the intuition behind each formal component without sacrificing rigor. In Section 4.2 and Section 5, we also illustrate the general process of rule extraction for MC and MC-max models.
>
> **Weakness 2**
> > Comparison with recent LLM-based reasoning models.
>
> **Response.**
>
> Our work focuses on models with *formal guarantee of faithfulness*, ensuring that the extracted rules exactly reproduce model predictions. In contrast, LLM-based methods lack such formal guarantee and may generate inconsistent or unverifiable results. Despite these differences, we have included representative LLM baselines for performance comparison. The results suggest that, in addition to providing formal guarantees, our models also achieve competitive performance.
>
> **Weakness 3**
> > The paper provides rule examples but no quantitative results; including such results could enhance the paper’s readability and help readers better understand the approach.
>
> **Response.**
>
> We have included Table 4 to report the number of extracted rules under different threshold $\gamma$, and we analysed the result in Section 6. The extracted rule sets are typically moderate in size (10–20 rules per head atom on average) indicating that they are feasible for human inspection and practical use.
>
> **Weakness 4**
> > The font issue.
>
> **Response.**
>
> This was caused by a misused imported package, which did not affect the font size or line spacing.
> We have fixed the issue in the current version.

---

### Official Review · Reviewer_4SqW · 2025-11-10

**Soundness:** 3
**Presentation:** 3
**Contribution:** 2
**Rating:** 4
**Confidence:** 3

**Summary:**

This paper develops two interpretable models (MC and MC-max) for tabular data cell completion that extract Datalog rules formally equivalent to the trained model's behavior. The MC model uses sum aggregation to count weighted paths between constants, while MC-max uses max aggregation, trading expressivity for extraction complexity. Theorems 4.4 and 5.2 prove that extracted rules produce identical predictions to the model on any input, guaranteeing explanation fidelity. Experiments on tabular (WP-IND, JF-IND, MFB-IND, FB-AUTO) and binary datasets show competitive empirical performance against HyperGCN, GMPNN, and LLM baselines, with rule extraction completing in 3-5 minutes. The main contribution is extending faithful rule learning from binary relations (knowledge graphs) to multi-arity relations (tables) with provable extraction algorithms.

**Strengths:**

-- This work appears to be the first faithful rule learning approach for multi-arity relations with provable extraction algorithms (Theorems 4.4, 5.2). Unlike approximate methods (Neural-LP, DRUM, SHAP), rules exactly replicate model behavior.

--  Max aggregation reduces extraction complexity with formal characterization of expressivity loss (Proposition E.1), addressing computational constraints.

-- Lemma 3.4 provides clear semantic interpretation as weighted path sums; multichain rules elegantly encode cardinality via inequalities; proofs are detailed with proper Datalog grounding.

-- Precise definitions distinguish cell completion from imputation and hypergraph prediction; comprehensive related work properly situates contribution within faithful rule learning and neuro-symbolic AI.

-- Rule extraction appears efficient on evaluation datasets given the reported extraction times; statistical significance testing validates improvements

**Weaknesses:**

-- Chain-like patterns cannot capture star patterns (simultaneous multi-attribute constraints), cycles, or branching logic—critical for real tables. Paper provides no analysis of what fraction of ground-truth relationships require non-chain patterns or comparison to unrestricted rule learners. Unclear when method is applicable vs. fundamentally limited.

-- The abstract claims "superior interpretability" but provides zero evidence beyond runtime and example rules. No rule complexity metrics, human studies, ground-truth comparison on synthetic data, or analysis of actionable rules. Core claim is unsubstantiated.

-- L=2, likely providing an explanation of extraction efficiency, and limiting practical effectiveness.

-- Neural-LP and DRUM are directly related but not compared. Both could potentially be adapted to multi-arity. Missing this baseline makes it impossible to evaluate whether formal guarantees provide practical advantages over approximate explanations.

**Questions:**

-- Can you provide examples of when L=2 rules provide impactful value to an application?  Can you provide L=3 or 4 examples (and analysis)?  What is the path length distribution in your dataset?  What are the practical limits for L?

-- Your LLM baselines show significant precision/recall imbalance suggesting implementation issues:
(a) Can you provide the CoT prompts used?
(b) What were TabLLM validation curves—did training succeed?
(c) Was temperature/top-p tuning attempted for calibration?
(d) Would more than 3-shot improve CoT?

-- Short chain-like patterns seem fundamentally limiting:  What fraction of relationships in your datasets require non-chain patterns (star, cycles, branching)? Can you provide synthetic experiments where ground-truth rules need complex structures?

---

> ### Author Response · Authors · 2025-11-20
> **Response for Reviewer 4SqW (Part 1)**
>
> We thank the reviewer for their valuable comments. We have incorporated the suggested clarifications in the updated paper.
>
> **Weakness 2**
> > The abstract claims "superior interpretability" but provides zero evidence beyond runtime and example rules.
>
> **Response.**
>
> Interpretability is a consequence of the formal equivalence between model predictions and the extracted Datalog rules: every prediction is exactly reproduced by applying the extracted rules to the input tables. Symbolic rules expose explicit predicates and variables, which are inherently easier to interpret than opaque neural parameters.
> Table 4 shows that rule extraction typically yields 10–20 rules per head predicate, making them practically inspectable. Concrete examples (Tables 5, 11, 12) suggest that the extracted rules often capture meaningful patterns such as symmetry, transitivity, and schema alignment.
>
> **Question 1**
> > Can you provide examples of when L=2 rules provide impactful value to an application? Can you provide L=3 or 4 examples (and analysis)? What is the path length distribution in your dataset? What are the practical limits for L?
>
> **Response.**
>
> Short rules (L=2) are both common and highly useful. They capture schema alignment across related tables—for example, in JF-IND, where the headers $\text{Player-Event-Team}$ and $\text{Team-Player-Event}$ lack explicit foreign keys,
> our extracted rules (Table 5) recover the missing alignments. Short rules also encode key logical dependencies; for instance, we were able to extract rules capturing symmetry (e.g., $\text{spouse}(x,y) \gets \text{spouse}(y,x)$), transitivity (e.g., $\text{also-see}(x,y) \gets \text{also-see}(x,z) \land \text{also-see}(z,y)$), and subsumption
> (e.g., $\text{subpart-of}(x,y) \gets \text{location-located-within-location}(x,y)$).
>
> Example rules for L=3 are provided and analysed in Appendix H. While longer chains can express more complex dependencies,  many are semantically implied by shorter rules. Additionally, many top-scored rules extracted under L=3 settings have actual chain lengths shorter than 3, indicating that the model effectively prunes redundant relational extensions.
>
> Most high-weighted rules have lengths $\leq 2$, while longer paths contribute marginal gains. Among the top-50 ranked rules extracted from WP-IND, JF-IND, MFB-IND, and FB-AUTO, 22.0\% and 27.5\% rules have lengths 1 and 2, respectively. As an example, the average length of rules extracted from FB-AUTO is 1.9 under the setting of L=3. Therefore, L=2 suffices for most practical dependencies.
>
> Beyond L=3, the number of possible path schemas grows exponentially with the number of predicates and their arities, increasing both computation cost and the risk of overfitting, resulting in worse performance. In practice, both L=2 and L=3 demonstrate a good balance between expressivity and efficiency. For larger hypergraph datasets, longer paths (e.g., L>3) can lead to training instability due to an explosion in parameters relative to data size, as we mentioned in Section 3.1.
>
> **Question 2**
> > Regarding LLM baselines: (a) Can you provide the CoT prompts used? (b) What were TabLLM validation curves—did training succeed? (c) Was temperature/top-p tuning attempted for calibration? (d) Would more than 3-shot improve CoT?
>
> **Response.**
>
> (a) Source codes (including prompts) can be found in the provided anonymous GitHub repository.
>
>
> (b) As analysed in Appendix H.1, fine-tuning causes the model to internalise spurious co-occurrences and frequent values, leading to many false positives. TabLLM’s training loss decreases slightly while validation loss shows little improvement. This reflects its design for independent row classification rather than the multi-hop relational reasoning required in our task.
>
>
> (c) Our experiment settings are detailed in Appendix G. We followed standard QLoRA fine-tuning settings without additional temperature or top-p calibration, as our primary goal was to broadly assess whether LLMs can handle the task.
>
>
> (d) We compared 3-shot and 5-shot CoT prompts. As reported in Table 1, 5-shot results slightly outperform 3-shot results on most datasets, but the improvement was not significant. Possible reasons include: (i) difficulty of capturing complex relational structure and reasoning patterns with few-shot text examples, and (ii) increase of the the input token length, which may distract the model's attention mechanism.

---

> ### Author Response · Authors · 2025-11-20
> **Response for Reviewer 4SqW (Part 2)**
>
> **Question 3**
> > Short chain-like patterns seem fundamentally limiting: What fraction of relationships in your datasets require non-chain patterns (star, cycles, branching)? Can you provide synthetic experiments where ground-truth rules need complex structures?
>
> **Response.**
>
> Our experiments show that a chain-centric design performs well on current benchmarks. However, as noted in Section 7, it cannot yet capture tree- or star-shaped dependencies. Handling such structures would require principled ways to select branching or cyclic expansions and existing benchmarks lack suitable examples, making systematic evaluation difficult.
> Extending the framework and developing new benchmark datasets are important directions for future work.
>
> **Weakness 4**
> > Comparison with Neural-LP models.
>
> **Response.**
>
> As validated by multiple prior works (see below), DRUM significantly outperforms Neural-LP on existing benchmarks. Since these models closely resemble each other, we opted for evaluating DRUM only.
>
> [1] M. Qu et al. RNNLogic: Learning Logic Rules for Reasoning on Knowledge Graphs. ICLR 2021
>
> [2] K. Cheng et al. Neural Compositional Rule Learning for Knowledge Graph Reasoning. ICLR 2023
>
> [3] X. Wang et al. Faithful Rule Extraction for Differentiable Rule Learning Models. ICLR 2024

---

### Author Response · Authors · 2025-12-01
**Summary of Updates**

We thank all the reviewers for their insightful feedback. We have clarified *all* the raised questions and updated the paper accordingly. Our main revisions include:

1. Added experiments under the $L \geq 3$ setting with detailed result analysis, and quantitative statistics of the extracted rules.
2. Included more illustrative examples to improve the ease of understanding.
3. Expanded the introduction to highlight practical application scenarios, and enriched the discussion of limitations and future work.

We believe these improvements have adequately addressed all the reviewers' concerns and strengthened the paper.

---

### Meta-Review · Area_Chair_XTd5 · 2026-01-06

**Summary:**

This paper tackles tabular cell completion with a strong interpretability goal. The authors propose two closely related models (MC and MC-max) that score candidate cell values via learned relational “paths” and then extract an equivalent set of Datalog rules. The key selling point is faithfulness: the extracted rules are meant to exactly match the model’s predictions, not just approximate or post-hoc explain them.

**Reviewer Concerns:**

Most concerns fall into three buckets. First, scope/expressiveness: the method is fundamentally centered on chain-like patterns, and reviewers worry this misses important table constraints (e.g., star/branching logic), with limited guidance on when this limitation matters in practice. Second, practicality at scale: global extraction for the MC variant can blow up, and both models are schema-bound, which may limit use in evolving database settings. Third, presentation and interpretability evidence: multiple reviewers found the paper hard to read, and initially wanted more concrete reporting on rule complexity/quantity and stronger contextual comparisons (including a broader view of table/LLM baselines).

**Reviewer Scores:**

Likely no change for the reviewers.

---

### Decision · Program_Chairs · 2026-01-26

Reject